# Learning from Both Structural and Textual Knowledge for Inductive Knowledge Graph Completion

**Kunxun Qi**[1]    **Jianfeng Du**[2,3,*]    **Hai Wan**[1,*]

[1] School of Computer Science and Engineering, Sun Yat-sen University, Guangzhou, China
[2] Guangzhou Key Laboratory of Multilingual Intelligent Processing,
Guangdong University of Foreign Studies, Guangzhou, China
[3] Bigmath Technology, Shenzhen, China
qikx@mail2.sysu.edu.cn, jfdu@gdufs.edu.cn, wanhai@mail.sysu.edu.cn

## Abstract

Learning rule-based systems plays a pivotal role in knowledge graph completion (KGC). Existing rule-based systems restrict the input of the system to structural knowledge only, which may omit some useful knowledge for reasoning, e.g., textual knowledge. In this paper, we propose a two-stage framework that imposes both structural and textual knowledge to learn rule-based systems. In the first stage, we compute a set of triples with confidence scores (called *soft triples*) from a text corpus by distant supervision, where a textual entailment model with multi-instance learning is exploited to estimate whether a given triple is entailed by a set of sentences. In the second stage, these soft triples are used to learn a rule-based model for KGC. To mitigate the negative impact of noise from soft triples, we propose a new formalism for rules to be learnt, named *text enhanced rules* or *TE-rules* for short. To effectively learn TE-rules, we propose a neural model that simulates the inference of TE-rules. We theoretically show that any set of TE-rules can always be interpreted by a certain parameter assignment of the neural model. We introduce three new datasets to evaluate the effectiveness of our method. Experimental results demonstrate that the introduction of soft triples and TE-rules results in significant performance improvements in inductive link prediction.

## 1 Introduction

Knowledge graph (KG) consists of real-world facts and has been widely used in many applications, including question answering [27], recommendation [22] and information retrieval [42]. A fact in KGs is usually represented by a *triple* of the form (*head*, *relation*, *tail*), where *head* and *tail* are entities. In general, KGs are highly incomplete since the complete set of facts is hard to collect. Therefore, *knowledge graph completion* (KGC), which aims to infer missing facts from observed ones, has become a vital ingredient for practically completing KGs.

In recent years, there are mainly two categories for prevalent approaches to KGC. One is the embedding-based category [3, 45, 33]. Methods in this category usually learn knowledge graph embeddings (KGE) by encoding entities and relations as low-dimensional real-value vectors. They have been shown to be effective for large-scale KGC, but are hard to generalize to the inductive scenario where missing facts involve previously unseen entities [20]. Besides, they can hardly be interpreted by human due to their black-box nature. The other category tackles KGC by learning rule-based systems, based on search algorithms with pruning heuristics [13, 24] or neural models for

---

*Corresponding authors

37th Conference on Neural Information Processing Systems (NeurIPS 2023).

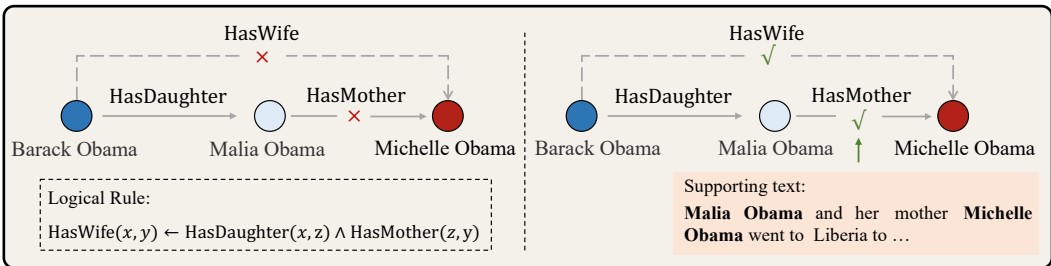

Figure 1: Reasoning from both structural and textual knowledge.

approximate rule learning [46, 30, 28]. They excel in explaining why a missing fact is inferred and are able to handle the inductive setting.

Previous work focuses merely on learning rule-based systems from structural knowledge. However, the high incompleteness of structural knowledge imposes a heavy burden for building high-quality rule-based systems. Text corpora can naturally be considered to enrich structural knowledge as they contain much additional knowledge for reasoning. Figure 1 showcases an example about how textual knowledge help to KGC. The sub-figure on the left side shows a logical rule to infer whether $x$ has wife $y$, and we cannot infer that "Barack Obama" has wife "Michelle Obam" since the key fact ("Malia Obama", HasMother, "Michelle Obam") is missing in the background KG. The sub-figure on the right side shows that text corpora can provide supporting texts to this key fact, thereby the target relation HasWife can be correctly inferred.

Motivated by this kind of examples, we aim to learn rule-based systems from both structural and textual knowledge. To this end, we propose a two-stage framework named *Learning from Structural and Textual Knowledge* or *LSTK* for short, as illustrated in Figure 2. In the first stage, a distantly-supervised method using a textual entailment model is proposed to extract textual knowledge from texts. Distant supervision [26] posits that if two entities involve a relation, then any sentences that mention these two entities might express that relation. Based on this assumption, we generate a set of triples with their mentioned texts and employ a textual entailment model equipped with a multi-instance learning mechanism to estimate whether a triple is entailed by a set of sentences. Each generated triple has an estimated confidence score denoting the supporting degree of the triple by texts, and thus it is called a *soft triple*. In the second stage, both the generated soft triples and hard triples (i.e. existing triples) are used to learn a rule-based system for KGC.

In practice, it is challenging to learn rules from soft triples, as the soft triples generated by distant supervision may contain noise. To mitigate the negative impact of noisy data, we propose a new formalism for rules to be learnt, named *text enhanced rules* or *TE-rules* for short. In this formalism, *textual relations* are introduced to provide a flexible mechanism to discard noisy soft triples. In other word, a TE-rule is extended from a chain-like rule by adding atoms or disjunction of atoms possibly with textual relations to the rule body. To effectively learn TE-rules, we propose a neural model named *TE-rule Learning Model* or *TELM* for short. We show that an arbitrary set of TE-rules can be encoded by a certain parameter assignment of TELM (see Theorem 1). This theoretical result guarantees a certain degree of faithfulness between TELM and TE-rules, enabling us to extract explainable TE-rules from the parameter assignment of the learnt TELM. Based on the extracted TE-rules, LSTK is able to provide explanations for inferred missing facts by backward reasoning.

We enhance three datasets from the field of relation extraction for empirical evaluation. Experimental results demonstrate significant gains achieve by the proposed method LSTK-TELM in inductive link prediction. Our ablation study and case study further clarify why the introduction of soft triples and TE-rules help to improve the performance.

## 2 Preliminaries

**Knowledge Graph.** Given a set of entities $\mathcal{E}$ and a set of relations $\mathcal{R}$, a knowledge graph $\mathcal{G}$ is a subset of $\mathcal{E} \times \mathcal{R} \times \mathcal{E}$. Specifically, $\mathcal{G} = \{(h_i, r_i, t_i)\}_{1 \leq i \leq N}$, where $N$ denotes the number of triples, $h_i \in \mathcal{E}$ the *head* entity for the $i^{\text{th}}$ triple, $r_i \in \mathcal{R}$ the relation for the $i^{\text{th}}$ triple and $t_i \in \mathcal{E}$ the *tail* entity for the $i^{\text{th}}$ triple. By $r^-$ we denote the inverse relation of $r \in \mathcal{R}$. The set of inverse relations for

$\mathcal{R}$, namely $\{r^- \mid r \in \mathcal{R}\}$, is denoted by $\mathcal{R}^-$. Accordingly, the equivalent knowledge graph for $\mathcal{G}$ composed by inverse relations, namely $\{(t, r^-, h) \mid (h, r, t) \in \mathcal{G}\}$, is denoted by $\mathcal{G}^-$.

**Logical Rule.** Most existing work focuses on learning chain-like rules (CRs). A CR is essentially a plain datalog rule [1] where all atoms are binary and every body atom shares variables with the previous atom and the next atom. Formally, a CR $R$ with $L$ body atoms, simply called an $L$-CR, is of the form:

$$H(x, y) \leftarrow B_1(x, z_1) \wedge B_2(z_1, z_2) \wedge ... \wedge B_L(z_{L-1}, y)$$

where $x$ is the head entity, $y$ the tail entity, and $z_1,\ldots,z_{L-1}$ variables. The part at the left (resp. right) side of $\leftarrow$ is called the *head* (resp. *body*) of $R$. The rule $R$ is called $r$-specific if $H = r$. By $H_R$ and $B_R$ we denote the atom in the head of $R$ and the set of atoms in the body of $R$, respectively. An atom or a rule is *ground* if it does not contain any variable. A rule $R$ is a fact if $B_R$ is empty and $H_R$ is ground. In this paper, a fact or a ground atom $r(a, b)$ and a *triple* $(a, r, b)$ are used interchangeably. To uniformly represent rules using fixed-length bodies, we introduce the *identity relation* (denoted by $I$) to rule bodies. For example, $r(x, y) \leftarrow s(x, y)$ can be converted into a rule with two body atoms, namely $r(x, y) \leftarrow s(x, y) \wedge I(y, y)$.

We say $\mathcal{G} \models H_R(a, b)$ if there exists a ground instance $R_g$ of $R$ such that $H_R(a, b) = H_{R_g}$ and $B_{R_g} \subseteq \mathcal{G} \cup \mathcal{G}^- \cup \{I(e, e) \mid e \in \mathcal{E}\}$. Let $\Sigma$ be a set of $r$-specific CRs and $(a, r, b)$ a triple. We say $\mathcal{G} \models_\Sigma (a, r, b)$ if there exits a logical rule $R \in \Sigma$ such that $\mathcal{G} \models H_R(a, b)$. We say a triple $(a, r, b)$ is *plausible* in $\mathcal{G}$ if there exists a set of possibly correct $r$-specific CRs $\Sigma$ such that $\mathcal{G} \models_\Sigma (a, r, b)$.

**Link Prediction.** Link prediction is the main task of knowledge graph completion that we focus on in this work. Given a knowledge graph $\mathcal{G}$, a head query $(?, r, t)$ or a tail query $(h, r, ?)$, the task of link prediction aims to find all entities $e \in \mathcal{E}$ such that $(e, r, t)$ for $(?, r, t)$ or $(h, r, e)$ for $(h, r, ?)$ is plausible in $\mathcal{G}$. The *inductive learning setting* requires that at least one entity in the test set should not appear in the training set, while relations in the test set also appear in the training set.

## 3 Methodology

In this section, we describe the proposed LSTK framework for inductive KGC with text corpus. We first formalize our problem setting as follows.

**Problem Setting.** Given a set of triples and a corpus for training $\mathcal{U}_{\text{train}} = (\mathcal{G}_{\text{train}}, \mathcal{T}_{\text{train}})$, a set of triples and a corpus for test $\mathcal{U}_{\text{test}} = (\mathcal{G}_{\text{test}}, \mathcal{T}_{\text{test}})$, our inductive KGC setting aims to learn a KGC system based on $\mathcal{U}_{\text{train}}$, and then evaluate the learnt system on $\mathcal{U}_{\text{test}}$. During evaluation, given a head query $(?, r, t)$ or a tail query $(h, r, ?)$, the learnt KGC system finds an answer with the highest estimated truth degree to answer this query, based on the background knowledge from $(\mathcal{G}_{\text{train}} \cup \mathcal{G}_{\text{test}} \setminus \{(h, r, t)\}, \mathcal{T}_{\text{train}} \cup \mathcal{T}_{\text{test}})$.

This setting is motivated by real-world application scenarios where we need to fetch texts from search engines to find evidences to verify a new fact. To address this problem setting, we propose *Learning from Structural and Textual Knowledge* or *LSTK* for short, as illustrated in Figure 2. Specifically, the proposed LSTK framework has two stages, where the first stage aims to generate soft triples from a text corpus and the second one aims at training a rule-based system for KGC. Furthermore, LSTK is able to provide an explanation to show why an inferred missing fact is plausible in the background knowledge, by tracking back the reasoning paths from both hard triples and soft triples, where soft triples can further be explained by tracking back the supporting texts.

### 3.1 Formalization of LSTK

In the first stage, our goal is to find all triples that are possibly entailed by the given text corpus. We achieve this goal by adopting distant supervision. As far as we know, distant supervision is mostly applied in relation extraction, where the relation between any two entities is restricted in a close set. However, we naturally hope to discover more open relations that can be entailed by the text corpus. Considering that textual entailment [31] can deal with open relations and is better at exploiting the semantic information of relational contexts, we impose distant supervision to a textual entailment model to generate soft triples.

**Distantly supervised data generation.** Given a knowledge graph $\mathcal{G}$ and a set of sentences (i.e. a text corpus) $\mathcal{T} = \{s_i\}_{1 \leq i \leq N_{\text{sen}}}$, we construct a training set $\mathcal{D}_{\text{train}} = \{(\tau_i, S_i, y_i)\}_{1 \leq i \leq N_{\text{tra}}}$ by

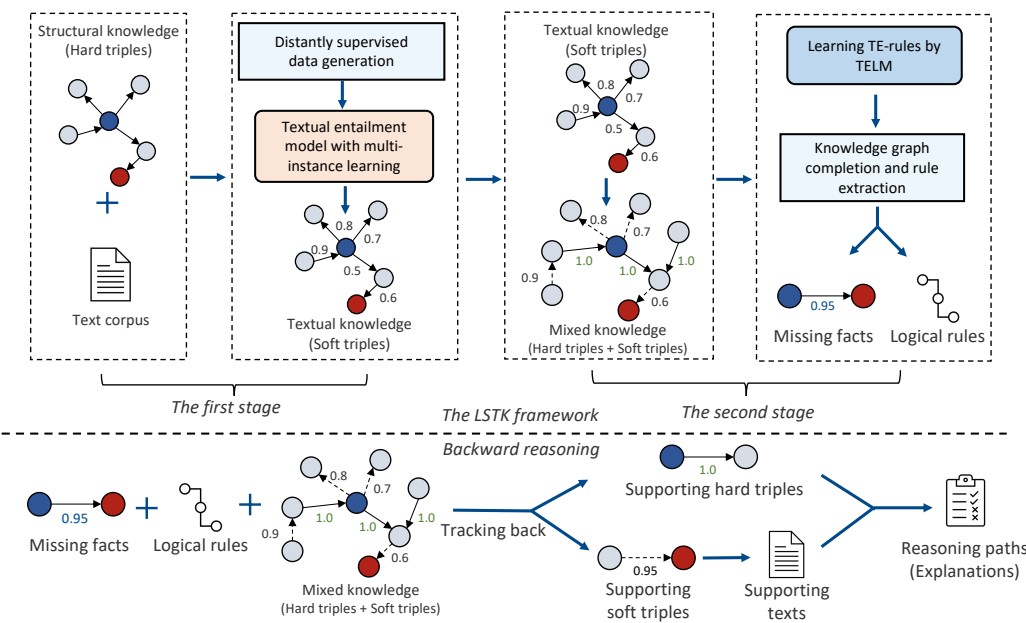

Figure 2: Overview of the proposed LSTK framework.

finding all sentences in $\mathcal{T}$ that mention the entity pair $(h_i, t_i)$ of $\tau_i$, where $N_{\text{sen}}$ denotes the number of sentences, $N_{\text{tra}}$ the number of training instances, $\tau_i = (h_i, r_i, t_i)$ a triple, $S_i$ the set of sentences in $\mathcal{T}$ mentioning the entity pair $(h_i, t_i)$, and $y_i$ the label. We call a triple $(h, r, t) \in \mathcal{G}$ a *positive triple*, whereas a triple $(h, r, t) \notin \mathcal{G}$ a *negative triple*. For each positive triple $(h_i, r_i, t_i)$, we append to the training set the set of negative triples $\{(h_i, r'_j, t_i) \mid r'_j \in \mathcal{R}, r'_j \neq r_i, (h_i, r'_j, t_i) \notin \mathcal{G}\}$ obtained by corrupting $r_i$. Accordingly $y_i$ is set to 1 if $\tau_i \in \mathcal{G}$ or 0 otherwise. Considering that some entity pairs may appear in a large number of sentences, to confine the number of considering sentences, we exploit the well-known graph partitioning algorithm METIS [17] to evenly partition the original $S_i$ to make every part no more than 20 sentences, and then treat the densest part defined by Jaccard similarity between bags of words as the ultimate $S_i$. Finally, we likewise construct the test set $\mathcal{D}_{\text{test}} = \{(\tau_i, S_i) \mid \tau_i \in \mathcal{E} \times \mathcal{R} \times \mathcal{E}, S_i \neq \emptyset\}$ for generating soft triples where labels are not given.

**Multi-instance learning for RTE.** Recognising textual entailment (RTE) [9, 4] aims at determining whether a sentence (called *hypothesis*) can be inferred by the other sentence (called *premise*). We employ a textual entailment model based on the pre-trained language model BERT [10] to estimate the confidence score for every triple. Specifically, given $\mathcal{D}_{\text{train}} = \{(\tau_i, S_i, y_i)\}_{1 \leq i \leq N_{\text{tra}}}$, we create a set of sequences $X_i = \{x_{i,j}\}_{1 \leq j \leq N_{S_i}}$ for each instance in $\mathcal{D}_{\text{train}}$ by filling the template "[CLS]<triple>[SEP]<sentence>[SEP]", where [CLS] and [SEP] are special tokens in BERT, <triple> denotes the slot to be filled by the context of $\tau_i$ and <sentence> the slot to be filled by the context of $s_j \in S_i$. All the sequences in $X_i$ are fed to BERT to calculate their contextual representations and the contextual embedding of the [CLS] token are used to calculate the entailment probability by a full-connected layer activated by the sigmoid function. By $\epsilon_{i,j}$ we denote the entailment probability for $x_{i,j}$, then the entire entailment probability $\phi_i$ for $X_i$ is calculated by $\phi_i = \max_{1 \leq j \leq N_{S_i}} \epsilon_{i,j}$. The entire model is trained by minimizing the cross-entropy loss.

**Soft triple generation.** The trained model is then applied to $\mathcal{D}_{\text{test}}$ for generating soft triples. A soft triple is defined as a quadruple $(h, r, t, \phi)$, where $(h, r, t)$ denotes a triple and $\phi$ its corresponding predicted entailment probability. Given a hyper-parameter $\delta$ to keep only highly confident soft triples, the set of soft triples that will be used in learning chain-like rules is defined as $\mathcal{G}_{\text{soft}} = \{(h_i, r_i, t_i, \phi_i) \mid (h_i, r_i, t_i) \in \mathcal{D}_{\text{test}}, \phi_i \geq \delta\}$.

### 3.2 Formalization of Text Enhanced Rules

In the second stage, we aim at learning logical rules from both hard (i.e. existing) and soft triples. However, it is challenging to learn rules from the background KG with soft triples, as the generated

soft triples by distant supervision may contain noise, which impairs the quality of learnt rules. The following example illustrates a case where noisy data impact the reasoning ability of a chain-like rule.

**Example 1.** *Considering the chain-like rule R in Figure 1:* $\text{HasWife}(x, y) \leftarrow \text{HasDaughter}(x, z) \wedge \text{HasMother}(z, y)$. *Suppose we have two soft triples "(A, HasDaughter, B, 0.55)" and "(B, HasMother, C, 0.90)" computed by the textual entailment model, where the former one is incorrect in the real world. From this rule R the negative triple "(A, HasWife, C)" will be incorrectly inferred to be positive.*

To alleviate the negative impact of noise from soft triples, a simple way is to restrict that some atoms in the rule should appear in the structural knowledge as existing triples. To this end, we extend chain-like rules by adding atoms or disjunctions of atoms that may involve *textual relations*. We call such extended chain-like rules *text enhanced rules* and simply TE-rules. Let $r_{\text{text}}$ denote the textual relation corresponding to the relation $r$, then a TE-rule can be defined below.

**Definition 1.** *A p-specific L-TE-rule R, simply a TE-rule if p and L are clear from the context, is of the form: $p(x, y) \leftarrow \mathcal{C}_1(x, z_1) \wedge \mathcal{C}_2(z_1, z_2) \wedge ... \wedge \mathcal{C}_L(z_{L-1}, y)$, where $\mathcal{C}_l$ can be an original relation, a textual relation, the identity relation, or a disjunction of an original relation and its corresponding textual relation, i.e., $\mathcal{C}_l(u, v)$ is of the form $r(u, v) \vee r_{\text{text}}(u, v)$.*

The following example explains why TE-rules can be used to resolve the error in Example 1.

**Example 2.** *Considering Example 1 again. By allowing TE-rules to be learnt, a learnt rule $R'$ can be: $\text{HasWife}(x, y) \leftarrow \text{HasDaughter}(x, z) \wedge (\text{HasMother}(z, y) \vee \text{HasMother}_{\text{text}}(z, y))$. From $R'$ the negative triple "(A, HasWife, C)" will not be inferred to be positive unless there exists a positive hard triple $(A, \text{HasDaughter}, B)$ for some B.*

## 3.3 End-to-end Learning of Text Enhanced Rules

To effectively learn TE-rules, we propose an end-to-end neural model named *TE-rule Learning Model* (*TELM* for short). The intuition of TELM is to select appropriate parameters to simulate the inference of TE-rules by gradient descent. Formally, given a set of hard triples $\mathcal{G}$, a set of soft triples $\mathcal{G}_{\text{soft}}$, the maximum number $L$ of body atoms in every rule, the maximum number of rules $N$, we first extend $\mathcal{G}$ to $\mathcal{G}_{\text{ext}} = \{(h, r, t, 1.0) | (h, r, t) \in \mathcal{G}\}$. Suppose $\mathcal{R} = \{r_i\}_{1 \leq i \leq n}$, its corresponding set of inverse relations $\mathcal{R}^- = \{r_i\}_{n+1 \leq i \leq 2n}$, its corresponding set of textual relations $\mathcal{R}_{\text{text}} = \{r_i\}_{2n+1 \leq i \leq 3n}$, its corresponding set of inverse textual relations $\mathcal{R}_{\text{text}}^- = \{r_i\}_{3n+1 \leq i \leq 4n}$, as well as $I = r_{4n+1}$. Let $\mathcal{K}_{\text{mix}} = \mathcal{G}_{\text{ext}} \cup \mathcal{G}_{\text{ext}}^- \cup \mathcal{G}_{\text{soft}} \cup \mathcal{G}_{\text{soft}}^- \cup \{(e, I, e, 1.0) \mid e \in \mathcal{E}\}$. For $1 \leq k \leq N, 1 \leq l \leq L$, TELM estimates a truth degree $\psi_{r,x,y}^{(k,l)}$ for each triple $(x, r, y) \in \mathcal{G}$, defined by:

$$\psi_{r,x,y}^{(k,l)} = \begin{cases} \sigma \left( \sum_{i=1}^{4n+1} \omega_i^{(r,k,L)} \mathbb{I}(\phi \mid (x, r_i, y, \phi) \in \mathcal{K}_{\text{mix}}) \right), & l = 1 \\ \sigma \left( \sum_{i=1}^{4n+1} \omega_i^{(r,k,L-l+1)} \sum_{z,\phi:(x,r_i,z,\phi) \in \mathcal{K}_{\text{mix}}} \psi_{r,z,y}^{(k,l-1)} \phi \right), & 2 \leq l \leq L \end{cases} \quad (1)$$

where $\sigma(x) = \max(\min(x, 1), 0)$ is an activation function that confines the given value to $[0, 1]$, $\mathbb{I}(\phi \mid C)$ is a function that returns $\phi$ if $C$ is true or 0 otherwise, and $\omega^{(r,k,l)} \in [0, 1]^{4n+1}$ denotes the trainable relational selection weights for the $l^{\text{th}}$ body atom of the $k^{\text{th}}$ rule for the head relation $r$. Intuitively, Equation (1) simulates the inference of TE-rules under the background KG with soft triples. The selection weights of all predicates for body atoms are formally defined as:

$$\omega^{(r,k,l)} = [\alpha^{(r,k,l)} w_{\text{orig}}^{(r,k,l)}; \beta^{(r,k,l)} w_{\text{text}}^{(r,k,l)}; w_{4n+1}^{(r,k,l)}] \quad (2)$$

where $[;]$ denotes the concatenation operator, and $w_{\text{orig}}^{(r,k,l)} \in [0, 1]^{2n}$ (resp. $w_{\text{text}}^{(r,k,l)} \in [0, 1]^{2n}$ or $w_{4n+1}^{(r,k,l)}] \in [0, 1]$) denotes the original relation selection weights (resp. textual relation selection weights or the identity relation selection weight) for the $l^{\text{th}}$ atom of the $k^{\text{th}}$ rule for the head relation $r$; further, $\alpha^{(r,k,l)} \in [0, 1]$ (resp. $\beta^{(r,k,l)} \in [0, 1]$) denotes the trainable selection weight for determining whether the original (resp. textual) relation is involved in the $l^{\text{th}}$ atom. Intuitively, $[w_{\text{orig}}^{(r,k,l)}]_i = 1$ (resp. $[w_{\text{text}}^{(r,k,l)}]_i = 1$ or $w_{4n+1}^{(r,k,l)} = 1$) denotes that the $i^{\text{th}}$ original relation $r_i$ (resp. the

$i^{\text{th}}$ textual relation $r_{2n+i}$ or the identity relation $r_{4n+1}$) is selected as the predicate of the $l^{\text{th}}$ atom. Note that $\alpha^{(r,k,l)} = 1 \wedge \beta^{(r,k,l)} = 1$ indicates that the disjunction of an atom with an original relation and an atom with a textual relation is involved in the $l^{\text{th}}$ atom. We confine $\alpha^{(r,k,l)}$ and $\beta^{(r,k,l)}$ to $[0,1]$ by sigmoid layers, normalize both $w_{\text{orig}}^{(r,k,l)}$ and $w_{\text{text}}^{(r,k,l)}$ by softmax layers, and restrict $w_{4n+1}^{(r,k,l)}$ to be $\max(0, 1 - \alpha^{(r,k,l)} - \beta^{(r,k,l)})$ so as to enforce $\alpha^{(r,k,l)} = \beta^{(r,k,l)} = 0$ when $w_{4n+1}^{(r,k,l)} = 1$.

The final predicted truth degree is calculated by a weighted sum of predicted degrees for $N$ rules:

$$\Psi_{r,x,y}^{(N,L)} = \sum_{k=1}^{N} u_r^{(k)} \psi_{r,x,y}^{(k,L)} \tag{3}$$

where $u_r^{(k)} \in [-1,1]$ is a trainable weight for the $k^{\text{th}}$ rule for the head relation $r$, which is confined to $[-1,1]$ by a tanh layer. Then the model is trained by minimizing the objective function

$$\mathcal{L} = - \sum_{(x,r,y) \in \mathcal{G}} \log \frac{\exp(\Psi_{r,x,y}^{(N,L)})}{\sum_{e \in \mathcal{E}} \exp(\Psi_{r,e,y}^{(N,L)})} \tag{4}$$

We then show the faithfulness between the formalization of TELM and TE-rules in Theorem 1, which relies on the notion of induced parameters as defined below.

**Definition 2.** *Given a set of $r$-specific $L$-TE-rules $\Sigma = \{R_k\}_{1 \leq k \leq N}$ for $R_k$ of the form $r(x,y) \leftarrow \mathcal{C}_{k,1}(x, z_1) \wedge ... \wedge \mathcal{C}_{k,L}(z_{L-1}, y)$ with $\mathcal{C}_{k,l}(u,v) \in \{r_{k,l}(u,v), r'_{k,l}(u,v), r^{\dagger}_{k,l}(u,v) \vee r^{\ddagger}_{k,l}(u,v)$, where $r_{k,l} \in \mathcal{R} \cup \mathcal{R}^- \cup \{I\}$, $r'_{k,l} \in \mathcal{R}_{\text{text}} \cup \mathcal{R}_{\text{text}}^-$, $r^{\dagger}_{k,l} \in \mathcal{R} \cup \mathcal{R}^-$, and $r^{\ddagger}_{k,l}$ is the corresponding textual relation of $r^{\dagger}_{k,l}$, we call a parameter assignment of TELM $\theta_r^{(N,L)} = \{[w_{\text{orig}}^{(r,k,l)}]_i, [w_{\text{text}}^{(r,k,l)}]_i\}_{1 \leq k \leq N, 1 \leq l \leq L, 1 \leq i \leq 2n} \cup \{w_{4n+1}^{(r,k,l)}, \alpha^{(r,k,l)}, \beta^{(r,k,l)}\}_{1 \leq k \leq N, 1 \leq l \leq L} \cup \{u_r^{(k)}\}_{1 \leq k \leq N}$ $\Sigma$-induced if it satisfies the following conditions for all $1 \leq k \leq N, 1 \leq l \leq L$:*

*(1) $\forall 1 \leq i \leq 2n : [w_{\text{orig}}^{(r,k,l)}]_i = 1$ if $r_i$ appears in $\mathcal{C}_{k,l}$, otherwise $[w_{\text{orig}}^{(r,k,l)}]_i = 0$.*

*(2) $\forall 1 \leq i \leq 2n : [w_{\text{text}}^{(r,k,l)}]_i = 1$ if $r_{2n+i}$ appears in $\mathcal{C}_{k,l}$, otherwise $[w_{\text{text}}^{(r,k,l)}]_i = 0$.*

*(3) $w_{4n+1}^{(r,k,l)} = 1$ if $r_{4n+1}$ appears in $\mathcal{C}_{k,l}$, otherwise $w_{4n+1}^{(r,k,l)} = 0$.*

*(4) $\alpha^{(r,k,l)} = 1$ if an original relation appears in $\mathcal{C}_{k,l}$, otherwise $\alpha^{(r,k,l)} = 0$.*

*(5) $\beta^{(r,k,l)} = 1$ if a textual relation appears in $\mathcal{C}_{k,l}$, otherwise $\beta^{(r,k,l)} = 0$.*

*(6) $u_r^{(k)} = 1$.*

**Theorem 1.** *Let $\mathcal{G}$ be a set of hard triples, $\mathcal{G}_{\text{soft}}$ a set of soft triples, $(a, r, b)$ an arbitrary triple, $\Sigma = \{R_k\}_{1 \leq k \leq N}$ a set of $r$-specific $L$-TE-rules and $\theta_r^{(N,L)}$ the $\Sigma$-induced parameter assignment of TELM, then $\Psi_{r,a,b}^{(N,L)} > 0$ if and only if $\mathcal{G} \cup \mathcal{G}_{\text{soft}} \models_\Sigma r(a,b)$.*

**Explaining by Backward Reasoning.** Thanks to the faithfulness between TELM and TE-rules, we can extract TE-rules by interpreting the parameters of a learnt TELM model using beam search algorithm (see Algorithm 1 in the appendix). For each inferred missing fact, LSTK yields at least one reasoning path as explanation by backward reasoning. Given a triple $(h, r, t)$ that is estimated to be positive, a set of extracted TE-rules, the background KG $\mathcal{K}_{\text{mix}}$ and the corpus $\mathcal{T}$, we can find a path between $h$ and $t$. Whenever the path involves a soft triple, we treat the sentence in $\mathcal{T}$ that has the maximum confidence score as the supporting text for this soft triple.

## 4 Experimental Evaluation

### 4.1 Data Construction and Evaluation Metrics

We collected three benchmark KGs with corresponding text corpora from the field of relation extraction for empirical evaluation, which are HacRED[2] [7], DocRED[3] [48] and BioRel[4] [41].

---

[2] https://github.com/qiaojiim/HacRED
[3] https://github.com/thunlp/DocRED
[4] https://bit.ly/biorel_dataset

| Dataset | #Ent. | #Rel. | #Train | #Valid | #Test | #Texts | #Soft. | PUE (exist) | PUE (soft) | PUE (all) |
|---------|-------|-------|--------|--------|-------|--------|--------|-------------|------------|-----------|
| HacRED | 20,800 | 12 | 20,637 | 2,499 | 2,551 | 4,578 | 38.8M | 98.7% | 31.6% | 30.0% |
| DocRED | 17,997 | 28 | 24,384 | 2,819 | 3,844 | 19,517 | 4.7M | 83.7% | 84.9% | 67.8% |
| BioRel | 15,566 | 34 | 25,854 | 2,406 | 3,032 | 150,687 | 123.4M | 79.4% | 62.1% | 46.8% |

Table 1: Statistical information for each dataset, where #Ent. (resp. #Rel.) denotes the number of entities (resp. relations), #Train/Valid/Test respectively the number of triples for training/validation/test, #Texts the number of texts and #Soft. the number of extracted soft triples. PUE (short for Proportion of Unseen Entities) denotes the proportion of triples in the test set involving unseen entities in the training triples, where exist/soft/all respectively denote that the training triples are treated as existing triples in the original training set, as extracted soft triples, or as both existing triples and soft triples.

Considering that these datasets originally have a number of relations that can hardly appear in heads of potential logical rules, we filtered these relations by applying AMIE+ [13] to mine logical rules that have up to three body atoms from the complete set of existing triples for each dataset and omitting the relations that have not been involved in the body atoms of any mined rules. Under the inductive setting, there should be entities in the test set that do not appear in the training set. Therefore, we applied the partitioning algorithm METIS [17] to the entity graph where an edge indicates that the two end-points have at least one filtered relation, so as to divide the set of entities into 10 subsets. The train/valid/test split of each dataset was then extracted from 8/1/1 of these subsets by omitting triples across different splits. Statistics on all above datasets are reported in Table 1.

Following [28], we reported the Mean Reciprocal Rank (MRR) and Hit@k metrics under the filtered setting [3] for empirical evaluation.

## 4.2 Implementation Details

We implemented the textual entailment model by Pytorch 1.10.0. The model was initialized by the pretrained language model BERT with 12 transformer layers, which outputs 768-dimensional (i.e. $d = 768$) token embeddings. Then it was trained by Adam [18] with warm up [10], where the initial learning rate was set to 5e-5, the mini-batch size to 8 and the maximum number of training epochs to 3. We applied dropout [32] to each layer by setting the dropout rate to 0.1.

We implemented TELM[5] by Pytorch 1.10.0. It was trained on an A100 GPU with 40GB memory by Adam [18] with 50 training epochs for HacRED and DocRED, and 20 for BioRel. The hyper-parameters were set to maximize MRR on the validation set. The initial learning rate was set to 1e-1 and the mini-batch size to 32 for all datasets. We also applied dropout [32] to the output states by setting the dropout rate to 0.3. The maximum length of each rule $L$ is set to 3 for HacRED and DocRED, and 2 for BioRel. The maximum size of learning rules $N$ is set to 20 for all datasets.

## 4.3 Main Results

We conducted experiments on the three enhanced datasets for link prediction under the inductive learning setting, where existing triples in the training set are used as background KG for training, existing triples in training and validation sets are used as background KG for validation, and existing triples in training, validation and test sets are used for test. The hyper-parameter $\delta$ to filter soft triples with high confidence scores was set to 0.5 in all experiments.

Table 2 reports the comparison results on three datasets. Textual Copy Rule (simply TCR) is a baseline that uses only *textual copy rules* of the form "$r(x, y) \leftarrow r_{\text{text}}(x, y)$" in reasoning instead of using any other rules or neural models. TCR has not any extra logical reasoning ability beyond the textual entailment model. By LSTK-X[†] we denote enhanced models that use soft triples in the background KG, where X denotes the original model. By LSTK-X we denote enhanced models that further extends LSTK-X[†] by allowing textual relations to appear in learnt rules. Results show that the proposed method (i.e. LSTK-TELM) significantly outperforms all the baseline methods with p-value < 0.05 by two-tailed t-tests. Specifically, LSTK-TELM outperforms LSTK-DRUM by absolute gains of 24.8%/4.2%/22.4% in Hit@1 scores on the HacRED/DocRED/BioRel datasets, respectively. Further, we can see from the comparison results between LSTK-X[†] and LSTK-X

---

[5]Code and data about our implementations are available at: `https://github.com/qikunxun/LSTK`

| Method | HacRED | | | | DocRED | | | | BioRel | | | | p-value |
|---|---|---|---|---|---|---|---|---|---|---|---|---|---|
| | MRR | H@1 | H@3 | H@10 | MRR | H@1 | H@3 | H@10 | MRR | H@1 | H@3 | H@10 | |
| Textual Copy Rule (TCR) | 0.382 | 31.5 | 42.2 | 51.5 | 0.030 | 1.8 | 3.6 | 5.4 | 0.074 | 4.5 | 8.2 | 13.4 | 3.3e-9 |
| AMIE+ [13] | 0.122 | 11.1 | 13.5 | 13.9 | 0.217 | 16.7 | 26.0 | 29.5 | 0.136 | 10.7 | 15.3 | 18.6 | 2.0e-7 |
| LSTK-AMIE+[†] | 0.177 | 13.3 | 19.0 | 28.2 | 0.259 | 21.4 | 29.1 | 33.4 | 0.138 | 10.2 | 15.0 | 20.2 | 1.9e-7 |
| LSTK-AMIE+ | 0.122 | 10.7 | 13.2 | 15.1 | 0.302 | 26.1 | 33.0 | 37.7 | 0.116 | 9.5 | 12.9 | 15.2 | 1.3e-3 |
| RNNLogic [28] | 0.162 | 15.1 | 17.2 | 17.8 | 0.226 | 15.7 | 26.3 | 35.6 | 0.172 | 12.0 | 19.3 | 28.7 | 1.9e-7 |
| LSTK-RNNLogic[†] | 0.234 | 16.6 | 26.2 | 37.3 | 0.343 | 24.9 | 40.4 | 52.1 | 0.148 | 10.0 | 16.0 | 23.2 | 5.8e-6 |
| LSTK-RNNLogic | 0.394 | 32.8 | 42.3 | 52.1 | 0.396 | 31.6 | 45.1 | 53.9 | 0.284 | 20.3 | 33.8 | 43.5 | 4.8e-6 |
| NeuralLP [46] | 0.395 | 34.4 | 42.6 | 48.9 | 0.310 | 24.0 | 35.4 | 43.4 | 0.317 | 23.2 | 26.8 | 46.2 | 1.3e-8 |
| LSTK-NeuralLP[†] | 0.320 | 21.1 | 35.4 | 56.7 | 0.382 | 29.2 | 43.5 | 55.2 | 0.304 | 21.2 | 34.5 | 47.2 | 1.6e-5 |
| LSTK-NeuralLP | 0.484 | 38.1 | 53.4 | 67.5 | 0.488 | 40.9 | 54.2 | 62.2 | 0.384 | 27.6 | 44.9 | 58.1 | 4.5e-4 |
| DRUM [30] | 0.387 | 33.6 | 41.4 | 48.5 | 0.352 | 28.1 | 40.2 | 47.7 | 0.314 | 23.1 | 36.2 | 46.4 | 3.0e-7 |
| LSTK-DRUM[†] | 0.347 | 23.8 | 39.6 | 56.0 | 0.392 | 29.1 | 46.3 | 56.5 | 0.292 | 20.4 | 32.9 | 46.0 | 1.5e-5 |
| LSTK-DRUM | 0.538 | 43.3 | 59.8 | 72.6 | 0.482 | 40.0 | 54.1 | 62.0 | 0.421 | 31.7 | 48.3 | 60.1 | 3.1e-4 |
| LSTK-TELM (this work) | **0.734** | **68.1** | **77.1** | 83.2 | **0.514** | **44.2** | **56.0** | **63.9** | **0.600** | **54.1** | **62.9** | **69.6** | - |
| (1) w/o soft triples | 0.416 | 37.7 | 43.6 | 48.1 | 0.380 | 30.8 | 44.0 | 49.4 | 0.339 | 25.9 | 38.3 | 48.1 | 1.1e-6 |
| (2) w/o textual copy rules | 0.710 | 65.0 | 74.9 | 82.1 | 0.502 | 43.2 | 54.6 | 62.7 | 0.511 | 44.6 | 54.7 | 62.8 | 2.1e-3 |
| (3) w/o textual relations | 0.579 | 48.2 | 64.3 | 76.4 | 0.438 | 35.5 | 49.1 | 58.5 | 0.356 | 25.4 | 40.3 | 54.8 | 5.1e-5 |
| (4) w/o disjunctive atoms | 0.693 | 63.3 | 73.2 | 80.6 | 0.453 | 38.4 | 49.6 | 57.7 | 0.430 | 36.3 | 45.7 | 55.6 | 2.8e-4 |
| (5) w/o confidence scores | 0.730 | 67.3 | 76.9 | 83.0 | 0.508 | 43.4 | 55.7 | 63.4 | 0.574 | 50.8 | 61.3 | 69.5 | 8.4e-3 |
| (6) Using relation extraction | 0.723 | 66.1 | 76.7 | **83.3** | 0.499 | 43.0 | 54.5 | 61.8 | 0.572 | 50.7 | 60.9 | 68.8 | 1.9e-4 |

Table 2: Comparison results on the HacRED, DocRED and BioRel datasets.

that, the introduction of textual relations bring significant performance improvements. For LSTK-NeuralLP[†] and LSTK-DRUM[†], their performances are even worse than their original models that do not use any soft triples, indicating some negative impact of noise from soft triples.

We further conducted ablation studies on several variants of LSTK-TELM to verify the effectiveness of key components in both LSTK and TELM. In (1) we omitted all the soft triples in the background KG (i.e., only hard triples are used). Results show that the LSTK framework (using soft triples) pushes TELM by a significant margin with p-value 1.1e-6. It implies that the use of soft triples help to greatly improve the performance. In (2) we omitted the corresponding soft triple for each hard triple in the background KG. Results show that this variant model already significantly outperforms the variant model in (1), indicating that soft triples help to improve the reasoning ability without using textual copy rules. In (3) we omitted the textual relations in TE-rules. Results show that the use of textual relations pushes LSTK-TELM by a significant margin with p-value 5.1e-5. It can be explained by the fact that the separation of original and textual relations helps to discard the noisy soft triples, resulting in higher reasoning performance. In (4) we omitted the learning of disjunctive atoms of the form $r(u, v) \vee r_{\text{text}}(u, v)$ in TELM. Results show that the introduction of disjunctive atoms bring significant performance gains with p-value 2.8e-4. This confirms that the introduction of disjunctive atoms in TE-rules provides a more flexible mechanism to control noise. In (5) we fixed the confidence scores (i.e. entailment probabilities) of soft triples to 1.0. Results show that the performance significantly drops with p-value 8.4e-3. It implies that the computed entailment probabilities reflect the truth degrees of supporting facts and help to learn more informative rules. In (6) we trained a state-of-the-art relation extraction model proposed in [2] based on BERT to replace the textual entailment model for generating soft triples. Results show that the use of relation extraction model leads to a significant performance degradation with p-value 1.9e-4. This may be due to that a textual entailment model can deal with open relations and exploit more text semantics on relations than a relation extraction model does.

## 4.4 Case Study

To clarify why the learning of TE-rules help improve the performance, we conducted case study for applying learnt logical rules from LSTK-TELM on the DocRED dataset, as shown in Table 3. In the first case, the logical rule is extracted from LSTK-TELM and can be applied to infer the positive triple "(Lois, Child, Superboy)". We can see from this case that the textual entailment model is able to infer from the relation mention "biological child" that, the entities "Lois" and "Clark" are spouse, thereby yielding a supporting soft triple "(Lois, Spouse, Clark, 0.76)" for reasoning. In the second case, it can be seen that the same logical rule in the first case cannot infer that the triple "(John, Child, Joseph)" is positive, since the key supporting fact "(Joseph, Father, Martyn)" is absent in the background KG. In the third case, the logical rule is "Child$(x, y) \leftarrow$ (Spouse$(x, z_0) \vee$ Spouse$_{\text{text}}(x, z_0)) \wedge$

| Triple (fact) | Logical rule | Supporting triple/text | Pred. | Label |
|---|---|---|---|---|
| (Lois, Child, Superboy) | $\text{Child}(x, y) \leftarrow (\text{Spouse}(x, z_0) \vee \text{Spouse}_{\text{text}}(x, z_0)) \wedge \text{Father}^-(z_0, y)$ | Supporting hard triple: (Superboy, Father, Clark) Supporting soft triple: (Lois, Spouse, Clark, 0.76) Supporting text: Clark and Lois' biological child in DC Comics canon was born in Convergence: Superman#2 (July 2015), a son named Jonathan Samuel Kent, who eventually becomes Superboy. | True | True |
| (John, Child, Joseph) | $\text{Child}(x, y) \leftarrow (\text{Spouse}(x, z_0) \vee \text{Spouse}_{\text{text}}(x, z_0)) \wedge \text{Father}^-(z_0, y)$ | Supporting hard triple: - Supporting soft triple: (John, Spouse, Martyn, 0.93) Supporting text: The Road to Ruin is a 1970 album released by husband and wife John and Beverley Martyn. | False | False |
| (Linda, Child, Joseph) | $\text{Child}(x, y) \leftarrow (\text{Spouse}(x, z_0) \vee \text{Spouse}_{\text{text}}(x, z_0)) \wedge \text{Father}^-_{\text{text}}(z_0, y)$ | Supporting soft triple: (Linda, Spouse, Paul, 0.88) Supporting text: In 1998, after Linda's death , Paul rearranged the song for string quartet to be played at memorial concerts for his late wife. Supporting soft triple: (Joseph, Father, Paul, 0.53) Supporting text: In 1995, at a ceremony in Colombo, Pope John Paul II beatified Father Joseph Vaz, an early missionary to the country, who is known as the Apostle of Ceylon. | True | False |

Table 3: Examples for applying learnt TE-rules from LSTK-TELM on the DocRED dataset, where Pred. abbreviates prediction. The words marked as blue denote entities and the words marked as red relation mentions. $r^-$ denotes the inverse relation of $r$.

$\text{Father}^-_{\text{text}}(z_0, y)$". In this case, the textual entailment model mistakenly yields a supporting soft triple "$(\text{Joseph}, \text{Father}, \text{Paul}, 0.53)$" based on the words "beatified Father", thereby wrongly inferring the triple "$(\text{Linda}, \text{Child}, \text{Joseph})$" to be positive. In general, these cases show that the use of soft triples is crucial in yielding a plausible explanation for a test triple and that the separation of original and textual relations provides a flexible mechanism to discard noisy soft triples.

## 5    Related Work

### 5.1    Knowledge Graph Completion

There are two categories of prominent approaches to knowledge graph completion. They are embedding-based and rule-based, respectively.

**Embedding-based methods.** Knowledge graph embeddings (KGE) [3, 39, 19, 45, 35, 33] are a kind of well-known embedding-based methods. They aim to represent entities and relations in KGs as low-dimensional real-valued vectors [37]. Although KGE methods have been shown to be effective in large-scale KGC, they can hardly generalize to the inductive learning setting. Besides, KGE methods are hard to be interpreted by human due to their black-box nature. In recent years, graph neural networks (GNNs) [34, 6, 23, 43, 50] have shown promising results in addressing KGC. They can also be considered as embedding-based, as they also learn latent representation for reasoning. While GNN-based methods have been shown to excel in handling the inductive learning setting, we have not incorporated GNNs with LSTK. The reasons are three-fold. Firstly, GNNs are still black-box models with limited interpretability, whereas the proposed LSTK framework is committed to explain why a missing fact is inferred. Secondly, GNN-based methods necessitate sub-graph extraction to handle test triples involving unseen entities. The process of sub-graph extraction becomes highly time-consuming when dealing with a large number of soft triplets, e.g., it takes about 10 hours for GraIL [34] to process 1M triples for sub-graph extraction. Thirdly, the confidence scores from soft triples cannot be directly incorporated in GNN-based methods, unless the message passing mechanism in GNN-based methods is redesigned to handle the confidence scores on links.

**Rule-based methods.** Different from embedding-based methods, rule-based methods aim at building effective rule-based systems for KGC. They can naturally reason under the inductive learning setting and provide logical rules as explanations. Learning rule-based systems has been widely studied in the field of Inductive Logic Programming (ILP). ILP methods such as AMIE+ [13] and AnyBURL [24] usually impose search algorithms with pruning heuristics to mine logical rules in a generate-and-test manner. RNNLogic [28] extends this manner to learn logical rules and their weights interactively based on neural models. In recent years, neural approximate methods are proposed to learn rule-based

systems from structural knowledge in KGs by learning continuous parameters using Tensorlog [8] operators. State-of-the-art neural approximate methods include NeuralLP [46], DRUM [30], NLIL [47] and Ruleformer [44]. In addition to applying Tensorlog, other neural approximate methods such as NTP [29] and CTP [25] learn logical rules based on neural theorem provers. All these methods learn rule-based systems from structural knowledge alone and most of them focus merely on learning chain-like rules. In contrast, we study learning more expressive TE-rules from both structural and textual knowledge.

**Other approaches using textual information.** It is worth noting that there are several approaches that focus on enhancing KGs with texts. For instance, recent advance of pre-trained language models (PLMs) for KGC such as KEPLER [38], PKGC [21] and SimKGC [36] have considered textual information of triples in KGs. The differences between PLM-based methods and LSTK are two-fold. On one hand, they consider different types of textual knowledge. In more detail, PLM-based methods consider text information such as contexts of triples, descriptions aligned to entities and pre-trained textual knowledge. In contrast, LSTK is proposed to address the KGC scenario where some knowledge (i.e., real-world facts) is provided in the structured KG and other knowledge is given by a text corpus. Thus, the textual knowledge considered in LSTK corresponds to a set of potential facts, namely the soft triples. On the other hand, PLM-based methods cannot work in our scenario because they can only obtain textual knowledge by pre-training a model over a given text corpus, whereas our setting allows to process new text corpora given in the test phase (see the problem setting in Section 3). The pre-training process is generally time-consuming and requires massive computing resources, thus it is impractical to be applied in the test phase. Except for PLM-based methods, the work [12] proposes a knowledge verification system by combining logical and textual information to compute explanations for validating new triples. However, the system should work with a predefined logical theory. In contrast, our approach does not require any predefined logical theory but learns one approximate logical theory on the fly. The recent work [15] improves the knowledge coverage of a knowledge base (KB) by facts extracted from text corpora. They have not focused on reasoning over facts. In contrast, we focus on enhancing the reasoning ability of rule-based systems by soft triples extracted from text corpora.

## 5.2 Distant Supervision for Relation Extraction

Distant supervision [26] is a dominant paradigm to handle the problem of lacking labeled data in relation extraction. It assumes that if two entities involve a relation, then any sentence mentioning them together might express that relation. Based on this assumption, a large number of labeled data for relation extraction can be automatically generated by aligning triples with texts. However, this may introduce noise to the generated data. Therefore, most previous studies exploit multi-instance learning [11] by putting instances with the same entity pair into bags, to alleviate the impact of noisy instances. A number of neural models including CNN [49], Bi-LSTM [5] and BERT [40] have been incorporated with multi-instance learning and used as classification models for relation extraction. We also impose distant supervision to score soft triples from a text corpus, but we resort to textual entailment models to make the best of relation mentions, instead of using classification models where relations are treated as labels to be predicted.

## 6   Conclusion and Future Work

In this work we have proposed a two-stage framework, named LSTK, to learn rule-based systems from both structural and textual knowledge. It computes a set of soft triples by distant supervision in the first stage and applies these soft triples to enhance the learning of neural approximate rule-based systems in the second stage. To mitigate the negative impact of noise from soft triples, we have proposed a new formalism for logical rules named TE-rules and a neural model named TELM for learning TE-rules. We introduced three new datasets for empirical evaluation. Experimental results demonstrate significant improvements achieved by learning TE-rules from soft triples. Our case study further reveals how TE-rules help to control noise from soft triples.

From the third case in case study, we observe that some errors produced by the textual entailment model will propagate to the final prediction. This is known as the error propagation issue in pipeline approaches. To tackle this issue, our future work will focus on studying fully end-to-end frameworks to learn TE-rules from both structural and textual knowledge.

## 7 Limitations

LSTK merely works for knowledge graphs with aligned text corpora, but in general knowledge graphs do not come with aligned texts. In order to adapt LSTK to more real-world applications, we generate soft triples automatically under the distant supervision assumption that any sentence mentioning two entities might involve a relation between these two entities. This assumption enables us to fetch texts via information retrieval tools such as search engines or to generate supporting text via large language models (LLMs) without the need of manually constructing texts. Note that all the original datasets of HacRED [7], DocRED [48] and BioRel [41] are collected by distant supervision. Although the performance of LSTK depends on the quality of distant supervision, LSTK indeed works for any knowledge graphs since aligned texts can easily be collected by distant supervision.

## 8 Acknowledgements

This paper was supported by National Natural Science Foundation of China (No. 62276284, 61976232, 61876204), National Key Research and Development Program of China (No. 2021YFA1000504), Guangdong Basic and Applied Basic Research Foundation (No. 2023A1515011470, 2022A1515011355, 2020A1515010642), Guangzhou Science and Technology Project (No. 202201011699), Shenzhen Science and Technology Program (KJZD20230923114059902), Guizhou Provincial Science and Technology Projects (No. 2022-259), Humanities and Social Science Research Project of Ministry of Education (No. 18YJCZH006), as well as the Fundamental Research Funds for the Central Universities, Sun Yat-sen University (No. 23ptpy31).

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

# A Proofs

## A.1 Proof of Lemma 1

To prove Theorem 1, we first introduce Lemma 1.

**Lemma 1.** *Let $\mathcal{G}$ be a set of hard triples, $\mathcal{G}_{\text{soft}}$ a set of soft triples, $(a, r, b)$ an arbitrary triple, $R$ an r-specific L-TE-rule of the form $r(x, y) \leftarrow \mathcal{C}_1(x, z_1) \wedge \mathcal{C}_2(z_1, z_2) \wedge ... \wedge \mathcal{C}_L(z_{L-1}, y)$, and $\theta_r^{(1,L)}$ the $\{R\}$-induced parameter assignment of TELM, then (1) $\psi_{r,a,b}^{(1,L)} > 0$ if $\mathcal{G} \cup \mathcal{G}_{\text{soft}} \models H_R(a, b)$, and (2) $\psi_{r,a,b}^{(1,L)} = 0$ if $\mathcal{G} \cup \mathcal{G}_{\text{soft}} \not\models H_R(a, b)$.*

*Proof.* Let $\mathcal{K} = \mathcal{G} \cup \mathcal{G}^- \cup \mathcal{G}_{\text{soft}} \cup \mathcal{G}_{\text{soft}}^- \cup \{(e, I, e) \mid e \in \mathcal{E}\}$ where $\mathcal{E}$ is the set of entities appearing in $\mathcal{G} \cup \mathcal{G}_{\text{soft}}$. Note that the confidence sore of each triple in $\mathcal{K}$ is larger than 0.

(I) Consider the case where $\mathcal{G} \cup \mathcal{G}_{\text{soft}} \models H_R(a, b)$. There exists at least one ground instance $R_g$ of $R$ such that $H_R(a, b) = H_{R_g}$ and $B_{R_g} \subseteq \mathcal{K}$. There will be a sequence of entities $c_1, \ldots, c_{L-1}$ and a sequence of relations $r_1, \ldots, r_L$ such that $(a, r_1, c_1), (c_1, r_2, c_2)\ldots, (c_{L-1}, r_L, b) \in \mathcal{K}$, where $r_i$ is a relation appearing in $\mathcal{C}_i$ for all $1 \leq i \leq L$. If $r_L \in \mathcal{R} \cup \mathcal{R}^-$, suppose $r_L$ is the $k^{\text{th}}$ relation in $\mathcal{R} \cup \mathcal{R}^-$, then by Condition 1 and 4 in Definition 2, we have $[w_{\text{orig}}^{(r,1,L)}]_k = 1$ and $\alpha^{(r,1,L)} = 1$. If $r_L \in \mathcal{R}_{\text{text}} \cup \mathcal{R}_{\text{text}}^-$, suppose $r_L$ is the $k^{\text{th}}$ relation in $\mathcal{R}_{\text{text}} \cup \mathcal{R}_{\text{text}}^-$, then by Condition 2 and 5 in Definition 2, we have $[w_{\text{text}}^{(r,1,L)}]_k = 1$ and $\beta^{(r,1,L)} = 1$. If $r_L = I$, by Condition 3 in Definition 2, we have $w_{4n+1}^{(r,1,L)} = 1$. By Equation (2), we must have $\omega_k^{(r,1,L)} = 1$ for some $k$. By Equation (1), we further have $\psi_{r,c_{L-1},b}^{(1,1)} \geq \phi_L > 0$ for $\phi_L$ the confidence score of $(c_{L-1}, r_L, b)$. Likewise, if $r_{L-1} \in \mathcal{R} \cup \mathcal{R}^-$, suppose $r_{L-1}$ is the $k^{\text{th}}$ relation in $\mathcal{R} \cup \mathcal{R}^-$, then by Condition 1 and 4 in Definition 2, we have $[w_{\text{orig}}^{(r,1,L-1)}]_k = 1$ and $\alpha^{(r,1,L)} = 1$. If $r_{L-1} \in \mathcal{R}_{\text{text}} \cup \mathcal{R}_{\text{text}}^-$, suppose $r_{L-1}$ is the $k^{\text{th}}$ relation in $\mathcal{R}_{\text{text}} \cup \mathcal{R}_{\text{text}}^-$, then by Condition 2 and 5 in Definition 2, we have $[w_{\text{text}}^{(r,1,L-1)}]_k = 1$ and $\beta^{(r,1,L-1)} = 1$. If $r_{L-1} = I$, by Condition 3 in Definition 2, we have $w_{4n+1}^{(r,1,L-1)} = 1$. By Equation (2), we must have $\omega_k^{(r,1,L-1)} = 1$ for some $k$. By Equation (1), we further have $\psi_{r,c_{L-2},b}^{(1,2)} > 0$. In the same way, we can show that $\psi_{r,c_{L-3},b}^{(1,3)} > 0, \ldots, \psi_{r,c_1,b}^{(1,L-1)} > 0$ and $\psi_{r,a,b}^{(1,L)} > 0$ in turn. Therefore, we have $\psi_{r,a,b}^{(1,L)} > 0$ if $\mathcal{G} \cup \mathcal{G}_{\text{soft}} \models H_R(a, b)$.

(II) Consider the case where $\mathcal{G} \cup \mathcal{G}_{\text{soft}} \not\models H_R(a, b)$. Suppose $\psi_{r,a,b}^{(1,L)} > 0$, then by Equation (1), there must be some $k \in \{1, \ldots, 4n + 1\}$ such that $\omega_k^{(r,1,1)} > 0$, $r_k$ appears in $\mathcal{C}_1$ and there exists $(a, r_k, c_1) \in \mathcal{K}$ fulfilling $\psi_{r,c_1,b}^{(1,L-1)} > 0$. Since $\psi_{r,c_1,b}^{(1,L-1)} > 0$, by Equation (1), there must be also some $l \in \{1, \ldots, 4n + 1\}$ such that $\omega_l^{(r,1,2)} > 0$, $r_l$ appears in $\mathcal{C}_2$ and there exists $(c_1, r_l, c_2) \in \mathcal{K}$ fulfilling $\psi_{r,c_2,b}^{(1,L-2)} > 0$. In the same way, we can show that there exists relation $r_m$ appearing in $\mathcal{C}_m$ and entity $c_m$ such that $(c_{m-1}, r_m, c_m) \in \mathcal{K}$ and $\psi_{r,c_m,b}^{(1,L-m)} > 0$ for $m = 3, \ldots, L - 1$ in turn, while there exists relation $r_L$ appearing in $\mathcal{C}_L$ such that $(c_{L-1}, r_L, b) \in \mathcal{K}$. Hence there exists a sequence of entities $c_1, \ldots, c_{L-1}$ and a sequence of relations $r_1, \ldots, r_L$ such that $(a, r_1, c_1), (c_1, r_2, c_2)\ldots, (c_{L-1}, r_L, b) \in \mathcal{K}$, where $r_i$ is a relation appearing in $\mathcal{C}_i$ for all $1 \leq i \leq L$. These two sequences constitute a ground instance $R_g$ of $R$ such that $H_R(a, b) = H_{R_g}$ and $B_{R_g} \subseteq \mathcal{K}$, contradicting $\mathcal{G} \cup \mathcal{G}_{\text{soft}} \not\models H_R(a, b)$. Thus $\psi_{r,a,b}^{(1,L)} \leq 0$. By Equation (1) we have $\psi_{r,a,b}^{(1,L)} \geq 0$. Therefore, we have $\psi_{r,a,b}^{(1,L)} = 0$ if $\mathcal{G} \cup \mathcal{G}_{\text{soft}} \not\models H_R(a, b)$. $\qquad\square$

## A.2 Proof of Theorem 1

*Proof.* Lemma 1 implies that, for all $R_k \in \Sigma$, $\psi_{r,a,b}^{(k,L)} > 0$ if $\mathcal{G} \cup \mathcal{G}_{\text{soft}} \models H_{R_k}(a, b)$ and $\psi_{r,a,b}^{(k,L)} = 0$ otherwise.

($\Rightarrow$) Suppose $\Psi_{r,a,b}^{(N,L)} > 0$. Then by Equation (3) and Condition 6 in Definition 2, there exists at least one r-specific L-TE-rule $R_k \in \Sigma$ such that $\psi_{r,a,b}^{(k,L)} > 0$. By Lemma 1 we have $\mathcal{G} \cup \mathcal{G}_{\text{soft}} \models H_{R_k}(a, b)$. Since $\mathcal{G} \cup \mathcal{G}_{\text{soft}} \models H_{R_k}(a, b)$ and $R_k \in \Sigma$, we have $\mathcal{G} \cup \mathcal{G}_{\text{soft}} \models_\Sigma (a, r, b)$.

($\Leftarrow$) Suppose $\mathcal{G} \cup \mathcal{G}_{\text{soft}} \models_\Sigma (a, r, b)$. Then we have $\mathcal{G} \cup \mathcal{G}_{\text{soft}} \models H_{R_k}(a, b)$ for some $R_k \in \Sigma$. By Lemma 1 we have $\psi_{r,a,b}^{(k,L)} > 0$ and for all $k' \neq k$, $\psi_{r,a,b}^{(k',L)} \geq 0$. By Equation (3) and Condition 6 in Definition 2, we have $\Psi_{r,a,b}^{(N,L)} > 0$. $\qquad\square$

## B    Formalization of Multi-instance Learning for RTE

In Section 4.1 we have described how to generate soft triples by applying a RTE model based on BERT [10]. Due to the space limitation, we omitted some details. Thus, in this section we further elaborate on how to apply multi-instance learning to RTE based on BERT.

Given $\mathcal{D}_{\text{train}} = \{(\tau_i, S_i, y_i)\}_{1 \leq i \leq N_{\text{tra}}}$, where $\tau_i = (h_i, r_i, t_i)$ denotes a triple, $h_i = (w_1^{h_i}, ..., w_{N_{h_i}}^{h_i})$ the head entity with $N_{h_i}$ tokens, $r_i = (w_1^{r_i}, ..., w_{N_{r_i}}^{r_i})$ the relation with $N_{r_i}$ tokens, $t_i = (w_1^{t_i}, ..., w_{N_{t_i}}^{t_i})$ the tail entity with $N_{t_i}$ tokens, $S_i = \{s_{i,j}\}_{1 \leq j \leq N_{S_i}}$ a set of $N_{S_i}$ sentences mentioning $(h_i, t_i)$, and $s_{i,j} = (w_1^{s_{i,j}}, ..., w_{N_{s_{i,j}}}^{s_{i,j}})$ a sentence with $N_{s_{i,j}}$ tokens, we create a set of sequences $X_i = \{x_{i,j}\}_{1 \leq j \leq N_{S_i}}$ for each instance in $\mathcal{D}_{\text{train}}$. For $1 \leq j \leq N_{S_i}$, the sequence $x_{i,j}$ is created by first concatenating $h_i, r_i, t_i$ and $s_{i,j}$, and then adding a special token [CLS] in front of the first token $w_1^{h_i}$, a special token [SEP] between $w_{N_{t_i}}^{t_i}$ and $w_1^{s_{i,j}}$, and a special token [SEP] behind the last token $w_{N_{s_{i,j}}}^{s_{i,j}}$. All sequences are fed to BERT to calculate their contextual representations. By $C_{i,j} \in \mathbb{R}^{N_{x_{i,j}} \times d}$ we denote the output of BERT for $x_{i,j}$, where $N_{x_{i,j}} = N_{h_i} + N_{r_i} + N_{t_i} + N_{s_{i,j}} + 3$ is the number of tokens in $x_{i,j}$ and $d$ is the dimension of token embedding. We use the embedding of the [CLS] token, namely $C_{i,j,1}$, as the contextual embedding for textual entailment classification. The entailment probability for $x_{i,j}$ is calculated by

$$\epsilon_{i,j} = \sigma(WC_{i,j,1} + b) \tag{5}$$

where $W \in \mathbb{R}^{1 \times d}$ is a trainable weight vector, $b \in \mathbb{R}^1$ a trainable bias value and $\sigma$ the sigmoid function. Then the entailment probability for $X_i$ is calculated by

$$\phi_i = \max_{1 \leq j \leq N_{S_i}} \epsilon_{i,j} \tag{6}$$

The entire model is trained by minimizing the cross-entropy loss, formally defined as

$$\mathcal{L} = -\frac{1}{N_{\text{tra}}} \sum_{i=1}^{N_{\text{tra}}} y_i \log \phi_i + (1 - y_i) \log(1 - \phi_i) \tag{7}$$

## C    Rule Extraction From TELM

Based on the formalization of TELM, we can interpret TE-rules from its parameters. The process of interpretation is shown in Algorithm 1. Intuitively, Algorithm 1 interprets TE-rules from the parameter assignment of TELM using beam search, where $\pi$ denotes the threshold to determine whether the original relations and textual relations are used in atoms, $m$ the beam size, $f_l$ the set of $(R', s)$-pairs for the $l^{\text{th}}$ atom, $R'$ the currently interpreted (partial) rule and $s$ its estimated score. It should be noted that the process for interpreting $r$-specific TE-rules outputs up to $m$ interpreted rules sharing the same confidence score $u_r^{(k)}$ for the $k^{\text{th}}$ target rule.

## D    Experiments

### D.1    Baselines

We applied LSTK to the following state-of-the-art rule learning methods for empirical comparisons.

- **AMIE+.** AMIE+ [13] is upgraded from the well-known rule learner AMIE [14]. It imposes several pruning heuristics to mine logical rules.
- **NeuralLP.** NeuralLP [46] is the first neural approximate method that exploits Tensorlog operators to learn chain-like rules. It translates the structure learning of logical rules from discrete space to continuous space.

---

**Algorithm 1:** Interpreting $r$-specific $L$-TE-rules

---

1 **Input:** beam size $m \geq 1$, selection threshold $\pi \in (0.0, 0.5]$, and a parameter assignment of
    TELM for the relation $r$, namely $\theta_r^{(N,L)} =$
    $\{[w_{\text{orig}}^{(r,k,l)}]_i, [w_{\text{text}}^{(r,k,l)}]_i\}_{1 \leq k \leq N, 1 \leq l \leq L, 1 \leq i \leq 2n} \cup \{w_{4n+1}^{(r,k,l)}, \alpha^{(r,k,l)}, \beta^{(r,k,l)}\}_{1 \leq k \leq N, 1 \leq l \leq L}$

2 **Output:** a set of up to $mN$ $r$-specific $L$-TE-rules

3 $\mathbb{R} \leftarrow \emptyset$;

4 **for** $1 \leq k \leq N$ **do**

5     $f_0 \leftarrow \{(\Delta^L, 1)\}$ where $\Delta$ denotes a placeholder to be filled;

6     $\forall 1 \leq l \leq L : f_l \leftarrow \emptyset$;

7     **for** $1 \leq l \leq L$ **do**

8         **for** $(R, s) \in f_{l-1}$ **do**

9             **if** $\alpha^{(r,k,l)} \geq \pi$ *and* $\beta^{(r,k,l)} < \pi$ **then**

10                 **for** $1 \leq i \leq 2n$ **do**

11                     $R' \leftarrow R$ with the $l^{\text{th}}$ placeholder replaced with $r_i$;

12                     $f_l \leftarrow f_l \cup \{(R', [w_{\text{orig}}^{(r,k,l)}]_i s)\}$;

13             **else if** $\alpha^{(r,k,l)} < \pi$ *and* $\beta^{(r,k,l)} \geq \pi$ **then**

14                 **for** $1 \leq i \leq 2n$ **do**

15                     $R' \leftarrow R$ with the $l^{\text{th}}$ placeholder replaced with $r_{2n+i}$;

16                     $f_l \leftarrow f_l \cup \{(R', [w_{\text{text}}^{(r,k,l)}]_i s)\}$;

17             **else if** $\alpha^{(r,k,l)} \geq \pi$ *and* $\beta^{(r,k,l)} \geq \pi$ **then**

18                 **for** $1 \leq i \leq 2n$ **do**

19                     $R' \leftarrow R$ with the $l^{\text{th}}$ placeholder replaced with $r_i \vee r_{2n+i}$;

20                     $f_l \leftarrow f_l \cup \{(R', \max([w_{\text{orig}}^{(r,k,l)}]_i, [w_{\text{text}}^{(r,k,l)}]_i) s)\}$;

21             **else**

22                 $R' \leftarrow R$ with the $l^{\text{th}}$ placeholder replaced with $I$;

23                 $f_l \leftarrow f_l \cup \{(R', w_{4n+1}^{(r,k,l)} s)\}$;

24         sort $f_l = \{(R, s)_j\}_{1 \leq j \leq N_{f_l}}$ in the descending order of $s$ and preserve the top-$m$ in $f_l$;

25     $\mathbb{Q} \leftarrow \{R' \text{ rewritten from } R \text{ to the form of a TE-rule} \mid (R, s) \in f_L\}$;

26     $\mathbb{R} \leftarrow \mathbb{R} \cup \mathbb{Q}$;

27 **return** $\mathbb{R}$;

---

- **DRUM.** DRUM [30] extends NeuralLP to learn rules with dynamic lengths by adding the identity relation. It exploits a bidirectional LSTM network to learn logical rules together with their scores simultaneously.

- **RNNLogic.** RNNLogic [28] leverages an LSTM [16] based rule generator and a reasoning predictor to learn logical rules together with their weights interactively.

For LSTK-NeuralLP, LSTK-DRUM and LSTK-RNNLogic, we constructed adjacency matrix for each relation from the mixed background $\mathcal{K}_{\text{mix}}$, where the weights of edges are set by the confidence scores. It should be noted that AMIE+ can only learn rules from hard triples in the background KG, and thus the confidence scores of soft triples are not used in LSTK-AMIE+.

### D.2 Implementation of Baselines

We implemented LSTK-NeuralLP[6] and LSTK-DRUM[7] by Tensorflow 1.15.0 based on their published code. They were trained by Adam, where the initial learning rate was set to 1e-3, the mini-batch size to 32, the maximum number of training epochs to 10, and the maximum length of learnt rules to 4.

---

[6]https://github.com/fanyangxyz/Neural-LP
[7]https://github.com/alisadeghian/DRUM

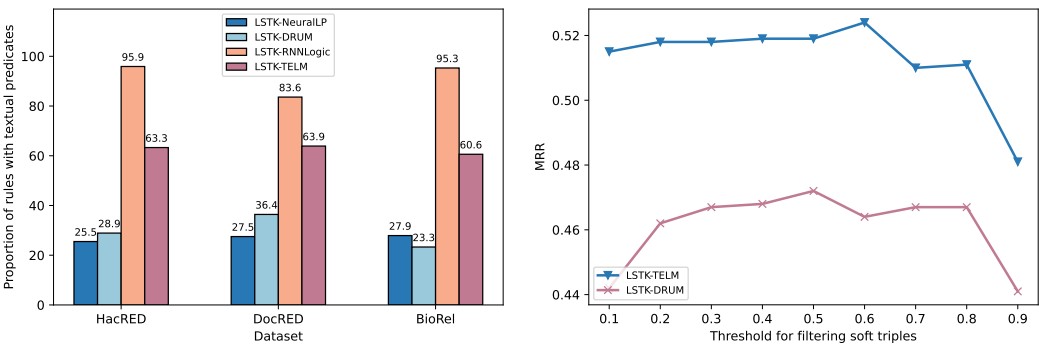

(a) Proportion of rules involving textual relations.

(b) Comparisons on $\delta$ for LSTK-TELM.

Figure 3: Analysis on learnt rules and the threshold $\delta$.

LSTK-RNNLogic[8] was implemented by Pytorch 1.10.0 based on its published code. Both the rule generator and reasoner in RNNLogic were trained by Adam, where the initial learning rate was set to 1e-3 (resp. 1e-1) for the generator (resp. the reasoner), the maximum number of training epochs to 5000 (resp. 5) for the generator (resp. the reasoner), the mini-batch size to 1024, and the maximum length of learnt rules to 2. Experiments on these baselines were conducted on an A100 GPU with 40GB memory. We implemented LSTK-AMIE+[9] based on its published code, and ran it on a Linux equipped with an Intel(R) Xeon(R) Gold 6248R processor with 3.0 GHz and 128 GB RAM.

### D.3 Evaluation Metrics

For each test triple $(h, r, t)$ in evaluation, we built two queries $(?, r, t)$ and $(?, r^-, h)$ for answering $h$ and $t$, respectively. For each query, we computed the truth degrees for corrupted head triples and then computed the rank of the correct answer. The rank of the correct answer is defined by $m + (n+1)/2$, where $m$ is the number of triples with higher truth degrees than the correct answer and $n$ the number of corrupted head triples with the same truth degree as the correct answer. The rank for each test triple is calculated by the mean rank of two queries. Based on the rank, we reported the Mean Reciprocal Rank (MRR for short) and Hit@k (H@k for short) metrics under the filtered setting.

### D.4 Statistical Analysis on Learnt Rules

We conduced statistical analysis on learnt rules to show how much textual knowledge has been used in different models, as shown in sub-figure (a) in Figure 3. We can see that about 30% logical rules extracted from either LSTK-NeuralLP or LSTK-DRUM (resp. about 60% logical rules extracted from LSTK-TELM) involve at least one textual relation. It implies that more soft triples are leveraged in the reasoning of LSTK-TELM. We have also found that about 90% logical rules extracted from LSTK-RNNLogic involve at least one textual relation, which is much higher than the rate of other approaches. This may be due to the large search space explored by RNNLogic in searching potential paths from the KG for rule generation, where the introduction of soft triples substantially scales up the search space, leading to a lot of logical rules that involve textual relations. Recall Table 2 that LSTK-RNNLogic achieves lower performance than other LSTK-enhanced models. These results imply that the overuse of textual relations may degrade the performance due to noise in soft triples.

### D.5 Hyper-parameter Analysis

LSTK introduces a threshold $\delta$ to filter soft triples with high confidence scores. We conducted experiments to show the impact of different setting of $\delta$, as shown in sub-figure (b) in Figure 3. Each curve on the MRR scores has a moderate jitter with increasing $\delta$. This can be explained by that, although the use of soft triples tends to improve the performance of link prediction, the overuse of soft triples may introduce much noise, leading to a performance degradation. By observing the curves of LSTK-DRUM and LSTK-TELM, we recommend to set $\delta = 0.5$ as default.

---

[8]https://github.com/DeepGraphLearning/RNNLogic

[9]https://www.mpi-inf.mpg.de/departments/databases-and-information-systems/research/yago-naga/amie/

