# OpenReview forum: "Learning from Both Structural and Textual Knowledge for Inductive Knowledge Graph Completion"
_NeurIPS.cc/2023/Conference — NeurIPS 2023 poster_

### Official Review · Reviewer_4GPq · 2023-07-03

**Soundness:** 4 excellent
**Presentation:** 3 good
**Contribution:** 3 good
**Rating:** 7
**Confidence:** 5

**Summary:**

This paper proposes an inductive knowledge graph completion method based on both textual and structural information. It introduces a new task of how to leverage information from both modalities when there are both structured and textual data available. This paper proposes a rule mining method called LSTK-TELM, where LSTK represents a rule mining framework. Three datasets are constructed based on the relation extraction dataset for evaluation. Experimental results demonstrate the effectiveness of the proposed framework, with LSTK-TELM outperforming other baseline methods.

**Strengths:**

S1. A novel knowledge graph completion scenario that combines both textual and RDF facts.

S2. This paper proposes a framework called LSTK to address this problem and introduces the LSTK-TELM method.

S3. Three datasets are constructed for evaluation. Experimental results demonstrate the effectiveness of the proposed method.

S4. Well written.

**Weaknesses:**

W1. There is a lack of discussion on related works in the field of inductive link prediction. See references below:
[1] Inductive Relation Prediction by Subgraph Reasoning
[2] Topology-aware correlations between relations for inductive link prediction in knowledge graphs
[3] Communicative Message Passing for Inductive Relation Reasoning
[4] Subgraph Neighboring Relations Infomax for Inductive Link Prediction on Knowledge Graphs
[5] Disconnected Emerging Knowledge Graph Oriented Inductive Link Prediction

W2. The use of the symbol "t" for both tail entities and rule lengths may lead to confusion.

**Questions:**

1. Rules can also be used for transductive knowledge graph completion. Why does this paper specifically mention focusing on inductive KGC? There doesn't seem to be any specific design for the inductive setting.

2. Please discuss the limitations of this paper. For example, the application scenario of the proposed task requires lots of text data related to the KG.

**Limitations:**

Please refer to the "Weaknesses" and "Questions" sections.

---

> ### Author Rebuttal · Authors · 2023-08-08
>
> Thanks for your helpful comments. The following addresses concerns and questions.
>
> > [R1] There is a lack of discussion on related works in the field of inductive link prediction.
>
> [A] Due to space limitations, we have only discussed the related work on learning rule-based systems for inductive link prediction. In reality, three main categories of methods have been proposed to address inductive link prediction, including GNN-based methods, PLM-based methods and rule-based methods. GNN-based methods such as R-GCN and other SOTA methods[1-5] address inductive link prediction based on graph neural networks (GNNs). PLM-based methods leverage pre-trained language models (PLMs) such as BERT to address the inductive link prediction problem. However, both GNN-based and PLM-based methods can be considered as embedding-based methods due to their black-box nature. For instance, the hidden representation learned by GNNs and PLMs can hardly be interpreted by human. Therefore, we focus on learning rule-based systems for inductive link prediction with textual knowledge, where logical rules (e.g., TE-rules) can serve as explanations for predictions. We naturally expect that the proposed framework LSTK is highly interpretable to human. Thus we have not incorporated GNN-based and PLM-based methods into LSTK for inductive KGC with textual knowledge. We will include these discussions in the revised version.
>
> > [R2] The use of the symbol "t" for both tail entities and rule lengths may lead to confusion.
>
> [A] Thank you for pointing out it. We will refine this notation in the revised version.
>
> > [R3] Rules can also be used for transductive knowledge graph completion. Why does this paper specifically mention focusing on inductive KGC? There doesn't seem to be any specific design for the inductive setting.
>
> [A] Inductive KGC is usually considered to be more practical and challenging in real-world scenarios. To guarantee the challenge and practicality of the proposed settings and datasets, we focus on inductive KGC in this work.
>
> > [R4] Please discuss the limitations of this paper. For example, the application scenario of the proposed task requires lots of text data related to the KG.
>
> [A] We have done this discussion in our supplement file (please see section E: Discussion on Limitations in the supplement file). We have discussed that the application scenario of our proposed framework requires KGs with corresponding text corpora. Considering that corresponding text corpora may be missing in most real-world KGs, we generate soft triples based on the distant supervision assumption that any sentence mentioning two entities might involve a relation between these entities. This assumption enables us to fetch texts via information retrieval tools such as search engines, without the need for manually filtering texts highly aligned with KGs. It is worth noting that both the orginal datasets of HacRED, DocRED and BioRel are collected using distant supervision. Considering that the process of fetching texts may introduce massive noises to the soft triples, we propose a novel formalism for logical rules (i.e. the TE-rules) to mitigate the negative impact of noises from soft triples.

---

> > ### Comment · Reviewer_4GPq · 2023-08-13
> >
> > The author's response has addressed most of my concerns. There is still one point to notice: the authors emphasize that GNN- and PLM-based link prediction is not human-interpretable and use this as a reason for not incorporating GNNs and PLMs. While interpretability is an advantage of rule-based methods, it is not a necessary requirement for inductive link prediction. Perhaps the authors should discuss them in light of the specificity of the proposed task, rather than interpretability.

---

> > > ### Author Response · Authors · 2023-08-15
> > > **Thanks for your helpful feedback!**
> > >
> > > Thank you again for taking the time to review our paper and providing valuable feedback. We will definitely improve our paper to clarify the relationships between related work and our work. We agree that discussing our work with GNN-based and PLM-based method in light of the specificity of inductive link prediction is necessary, and we will include these discussions in the revised version. Specifically, in the response to reviewer Vj83, we have outlined the reasons why PLM-based methods are not suitable to be applied in our inductive settings. Regarding GNN-based methods, rather than prioritizing interpretability, we will compare them to our work from two other perspectives.
> > >
> > > On one hand, GNN-based methods [1-5] necessitate subgraph extraction to handle test triples involving unseen entities. The process of subgraph extraction becomes highly time-consuming when dealing with a large number of soft triplets (e.g., it takes about 10 hours for GraIL [1] to process 1M triples for subgraph extraction). This limitation impairs their practicality in our inductive scenario.
> > >
> > > On the other hand, our link prediction setting comprises two types of triples, namely the hard triples and soft triples. In LSTK, we employ the confidence scores of soft triples to capture the uncertainty of potential facts derived from supporting texts. However, these confidence scores cannot be directly incorporated in GNN-based methods, as they require newly designed message passing function (aggregate function) to capture such features.
> > >
> > > Thank you for the helpful discussion for improving our work again! Please feel free to provide additional feedback and we will try our best to improve our work!

---

> > > > ### Comment · Reviewer_4GPq · 2023-08-17
> > > >
> > > > Thanks. I don't have further comments.

---

### Official Review · Reviewer_PLPA · 2023-07-04

**Soundness:** 3 good
**Presentation:** 3 good
**Contribution:** 2 fair
**Rating:** 6
**Confidence:** 3

**Summary:**

This paper proposes a two-stage framework that imposes both structural and textual knowledge to conduct knowledge graph completion. The first stage aims to extract some soft triples with confidence scores. And then the second stage designs some tailored rules for entity link prediction, including text-enhanced rules and TE-rules. Experiments on three benchmarks show the proposed method significantly outperforms previous work.

**Strengths:**

1. The idea of this paper is clear.
2. Experimental results show significant improvements to baseline models.

**Weaknesses:**

1. The text-enhanced entity representation models have been widely studied, such as [1][2]. Related work should be discussed.
2. The two-step pipeline will bring unnecessary cascaded errors. The most directed way is enhancing the entity representations using text representations to directly predict the entity links.
3. The first step with confidence score seems to model uncertainty in linking entities, which may be a novelty. However, such a motivation is not clearly described.
4. The relationship between the designed step should also be emphasized. Such as, the first step brings more entity semantics from text and the second step verifies the entity relations via text and structure aspects.
5. LLMs have shown strong ability in building KGs and exacting entity relations. This paper does not conduct experiments on LLMs.

[1] Representation Learning of Knowledge Graphs with Entity Descriptions. AAAI 2016.
[2] Entity-duet neural ranking: Understanding the role of knowledge graph semantics in neural information retrieval. ACL 2018.

**Questions:**

The motivations of the designed two step is unclear.

---

> ### Author Rebuttal · Authors · 2023-08-08
>
> Thanks for your helpful comments. The following addresses concerns and questions.
>
> > [R1] The text-enhanced entity representation models have been widely studied, such as [1][2]. Related work should be discussed
>
> [A] Due to space limitations, we have not discussed the text-enhanced entity representation models such as [1-2]. These methods follow a different research line compared to our method. The methods proposed in [1-2] focus on enhancing entity representation by text such as the descriptions on entities, relying on aligned text descriptions on entities as input. In contrast, we study the KGC setting where a part of the knowledge (i.e. real-world facts) exists in a structured KG, while the other part exists in text corpus. Thus no descriptions on entities are given in our setting. We require to first extract textual knowledge (i.e., a set of soft triples) from the given text corpus and then apply these soft triples to learn rule-based systems for inductive KGC. We will include these discussions in the revised version.
>
> > [R2] The two-step pipeline will bring unnecessary cascaded errors. The most directed way is enhancing the entity representations using text representations to directly predict the entity links.
>
> [A] The mentioned way is infeasible in the scenario considered in this paper. On one hand, we study learning rule-based systems from both structural and textual knowledge, where there is no entity representation in a rule-based system. We have not studied embedding-based methods in LSTK because they are black-box models with low interpretability, and can hardly generalize to the inductive setting. We naturally expect that the proposed framework LSTK is highly interpretable to human. On the other hand, the text-enhanced entity representation models assume that the manual annotated descriptions on entities are explicitly given, whereas the inputs of our setting are a KG and its corresponding text corpus. Thus there are no text descriptions on entities for enhancing entity representations. In our setting, we assume that a part of the knowledge (i.e. real-world facts) exists in structured KG, while the other part exists in text corpus. Therefore, a two-step pipeline is necessary in LSTK to first extract potential facts (i.e. the soft triples) from the given text corpus, and then apply these soft triples to learn rule-based systems.
>
> > [R3] The first step with confidence score seems to model uncertainty in linking entities, which may be a novelty. However, such a motivation is not clearly described.
>
> [A] The goal of the first stage in LSTK is not to link entities but to mine a set of potential facts (i.e. the soft triples) for learning rule-based systems. In this work, we assume that some knowledge exists in the given KG and some exists in the given text corpus. The motivation of the first stage in LSTK is to discover textual knowledge in the text corpus by mining soft triples. This process is not for linking entities, but for relation extraction from every entity-pair that is mentioned by texts.
>
> > [R4] The relationship between the designed step should also be emphasized. Such as, the first step brings more entity semantics from text and the second step verifies the entity relations via text and structure aspects.
>
> [A] The goal of the first step is not to brings more entity semantics from text but to mine potential knowledge (i.e. real-world facts) from the text corpus. The goal of the second step is to learn a rule-based system for addressing inductive KBC, using both existing hard triples and extracted soft triples. We have describe the relationship between these two step in introduction (line 44-52) and methodology (line 132-134).
>
> > [R5] LLMs have shown strong ability in building KGs and exacting entity relations. This paper does not conduct experiments on LLMs.
>
> [A] To the best of our knowledge, no published work has shown that LLMs such as ChatGPT and GPT-4 excel at addressing inductive KGC. At the present stage, LLMs are unsuitable for adoption in our scenario. The reasons are two-fold. First, applying LLMs in the first stage of LSTK is heavy in cost, as the first stage requires repeatedly applying the relationship extraction model. For instance, more than 100 million soft triples are extracted from the BioRel dataset (please see Table 1), which means that we require extensive invocation of LLMs. However, the number of requests for LLMs is still limited to this day. Second, it is infeasible to apply LLMs such as ChatGPT and GPT-4 in the second stage for inductive KGC because we cannot directly fine-tune LLMs for our task due to the resource limitations and closed source limitations. One solution may be applying LLMs using few-shot learning or in-context learning, but this solution may also suffer from the problems of low reasoning quality and length limitation when handling long context (e.g., the context of the entire KG is required). This is also out of our research field because LSTK focuses on providing faithful explanations for inductive KGC, based on a rule-based system learned from both structural and textual knowledge.
>
> > [R6] The motivations of the designed two step is unclear.
>
> [A] In our settings, we assume that a part of the knowledge exists in a structured KG, while the other part exists in text corpus. To mine textual knowledge (i.e., potential facts) from the given text corpus, we first employ a textual entailment model to generate soft triples. In the second stage, the generated soft triples are used to learn a rule-based system for inductive KGC. Considering that the number of mined soft triples is large, we cannot optimize these two-step in an end-to-end manner, as the end-to-end manner requires to store the gradients of all soft triples, resulting in out-of-memory (OOM) issues. Thus, we design a two-step framework in LSTK.

---

> > ### Comment · Reviewer_PLPA · 2023-08-17
> >
> > Thank you for your explanation. My concerns have been addressed. Is there any possible way to conduct some signals from the rule based systems to optimize the KG-sentence entailment method, such as RL?  The rule based systems seems a expert that contains lots of logical rule knowledge.

---

> > > ### Author Response · Authors · 2023-08-18
> > > **Thanks for your helpful feedback!**
> > >
> > > Thank you for taking the time to review our paper and providing helpful and valuable feedback. We will definitely improve our paper based on your comments. Furthermore, your additional feedback regarding the integration of signals from rule-based systems to enhance the KG-sentence entailment method with optimizing tactics such as RL is quite interesting and meaningful. Actually, we have been committed to examining such proposals, including optimization tactics like RL and gradient grafting [1]. The primary challenge we are facing is the issue of training efficiency, as there are more than 100 million soft triples that need to be used for enhancement in some datasets. This may require some well-designed heuristic sampling strategies to reduce the search space. Considering that the proposed two-step framework (LSTK) is efficient and effective enough (e.g., all MRR scores exceed 0.5 by LSTK-TELM) to handle soft triples, we leave this investigation to future work. We have also mentioned this in our discussion on future work in the conclusion, and we will include more details in the revised version. Thank you for the helpful discussion for improving our work again. Please feel free to provide additional feedback and we will try our best to improve our work further!
> > >
> > > [1] Wang Z, Zhang W, Liu N, Wang J. Scalable Rule-Based Representation Learning for Interpretable Classification. In NeurIPS 2021: 30479-30491.

---

> > > > ### Comment · Reviewer_PLPA · 2023-08-18
> > > >
> > > > Thank you for your response. It is really wonderful to add more experiments and tell all experiments clear. I increase my score to weak accept. I wish more further work can by conducted. Thank you for the hard working of all authors.

---

> > > > > ### Author Response · Authors · 2023-08-19
> > > > >
> > > > > Thank you for taking the time to discuss our paper and provide valuable insights to further improve our work. Thanks again!

---

> > ### Comment · Reviewer_PLPA · 2023-08-17
> >
> > I should cite the KEPLER and BERTRL, which represent entity representations in w1. I rethink it and maybe direct triple is enough, because it is modeled as a sentence entailment task and the entity description is long and may brings additional noise. But as shown in other review, such a point can be discussed.

---

> > > ### Author Response · Authors · 2023-08-18
> > > **Thanks for your further feedback!**
> > >
> > > Thanks for your further valuable comments. We agree that the reason why we did not introduce entity descriptions as input in our settings should be further discussed. As you mentioned, the entity description is long and may bring additional noise. This should be one of the reasons. Apart from this, we consider two more reasons. Firstly, having an aligned entity description for each entity may impair the practicality, because the description may be missing for some entities in real-world scenarios, especially for certain vertical domain KGs like medical KGs. Secondly, there may be multiple descriptions (expressing different meaning) for some entities, which requires an effective mechanism to incorporate such signals into logical rules. Otherwise, it might introduce more noise into the reasoning process. We will include these discussions in the revised version. Thanks again!

---

### Official Review · Reviewer_Vj83 · 2023-07-06

**Soundness:** 3 good
**Presentation:** 3 good
**Contribution:** 3 good
**Rating:** 6
**Confidence:** 3

**Summary:**

In this paper, the authors try to solve Knowledge Graph Completion (KGC) by proposing a two-stage framework Learning from Structural and Textual Knowledge (LSTK), that imposes both structural and textual knowledge to learn rule-based systems. In the first stage, the authors compute a set of triples with confidence scores. To mitigate the influence of the noise in the set of triples estimated in the first stage, the authors also propose a threshold-based method text enhanced rules (TE-rules) to filter out low-confident rules. In the second stage, the authors use the filtered triples by TE-rules to train a neural model named TE-rule Learning Model (TELM) for KGC. Furthermore, the authors created new datasets based on the data, HacRED, DocRED, and BioRel. Experimental results show that the proposed method LSTK outperforms the baseline methods regarding MRR, Hits@k (k=1,3,10).

**Strengths:**

- Since the proposed method LSTK relies on rule-based inference, the estimation of this method is interpretable to humans.
- The proposed approach can restrict the automatically extracted rules to high-confidence ones by TE-rules to deal with the noisy rules extracted by distant supervision.
- The threshold of filtering soft triples does not require sensitive tuning and works with 0.5. Thus, LSTK is robust and easy to use.
- This work provides new KG datasets with their corresponding texts.

**Weaknesses:**

- The paper does not refer to the recent KGC model that can conduct KGC with unseen entities [1].
- Even though recent KGC models utilize pretrained language models considering textual information of triplets in KGs [2], there is no comparison against such models.

[1] Wang, X., Gao, T., Zhu, Z., Zhang, Z., Liu, Z., Li, J., & Tang, J. (2021). KEPLER: A unified model for knowledge embedding and pre-trained language representation. Transactions of the Association for Computational Linguistics, 9, 176-194. (https://direct.mit.edu/tacl/article/doi/10.1162/tacl_a_00360/98089/KEPLER-A-Unified-Model-for-Knowledge-Embedding-and)

[2] Lv, X., Lin, Y., Cao, Y., Hou, L., Li, J., Liu, Z., ... & Zhou, J. (2022, May). Do Pre-trained Models Benefit Knowledge Graph Completion? A Reliable Evaluation and a Reasonable Approach. In Findings of the Association for Computational Linguistics: ACL 2022 (pp. 3570-3581). (https://aclanthology.org/2022.findings-acl.282/)

**Questions:**

As shown in the example of Figure 1, the proposed method LSTK can cover inferences of unseen entities and relations. However, as shown in the inductive setting of Wikidata5M [1], recent prompt-based KGC models can complete such triples based on text information like descriptions and pretrained knowledge similar to your work. Thus, to claim the advantage of LSTK, you need to compare it to the recent KGC models like KEPLER [1] and SimKGC [3]. Also, could you explain the difference between these models and LSTK?

[1] Wang, X., Gao, T., Zhu, Z., Zhang, Z., Liu, Z., Li, J., & Tang, J. (2021). KEPLER: A unified model for knowledge embedding and pre-trained language representation. Transactions of the Association for Computational Linguistics, 9, 176-194. (https://direct.mit.edu/tacl/article/doi/10.1162/tacl_a_00360/98089/KEPLER-A-Unified-Model-for-Knowledge-Embedding-and)

[3] Wang, L., Zhao, W., Wei, Z., & Liu, J. (2022, May). SimKGC: Simple Contrastive Knowledge Graph Completion with Pre-trained Language Models. In Proceedings of the 60th Annual Meeting of the Association for Computational Linguistics (Volume 1: Long Papers) (pp. 4281-4294). (https://aclanthology.org/2022.acl-long.295/)

**Limitations:**

I think you need to add that this approach is difficult to work on knowledge graphs that do not have corresponding text data to Limitations in your paper.

---

> ### Author Rebuttal · Authors · 2023-08-08
>
> Thanks for your helpful comments. The following addresses concerns and questions.
>
> > [R1] The paper does not refer to the recent KGC model that can conduct KGC with unseen entities [1].
>
> [A] Due to space limitations, we have only discussed the related work about learning rule-based systems for inductive KGC. Up to now, three main categories of methods have been proposed to address KGC, including GNN-based methods, PLM-based methods and rule-based methods. GNN-based methods such as R-GCN address inductive link prediction based on graph neural networks (GNNs). PLM-based methods such as [1-3] leverage pre-trained language models (PLMs) such as BERT to address the inductive KGC problem. However, both GNN-based and PLM-based methods can be considered as embedding-based methods due to their black-box nature. For example, the hidden representation learned by GNNs and PLMs can hardly be interpreted by human. Therefore, we focus on learning rule-based systems for inductive KGC with textual knowledge, where logical rules (e.g., TE-rules) can serve as explanations for predictions. We will include these discussions in the revised version.
>
> > [R2] Even though recent KGC models utilize pretrained language models considering textual information of triplets in KGs [2], there is no comparison against such models.
>
> [A] The approach like [2] follows a different research line than LSTK, considering distinct textual knowledge from that of LSTK. The textual knowledge considered in the work [2] encompasses the contexts of triples, descriptions on entities and the pre-trained knowledge. In contrast, we study the KGC setting where a part of the knowledge exists in the KG, while the other part (i.e. potential facts) exists in text corpus. In this setting, the input consists solely of the provided KG and its corresponding text corpus, without any accompanying entity descriptions. The textual knowledge considered in our work pertains to potential facts entailed by the text corpus. Consequently, we mine soft triples from the text corpus in the first stage. We focus on learning rule-based system because logical rules excel in explaining why a missing fact is inferred. While PLMs are capable of handling inductive KGC, they remain black-box models with limited interpretability, and cannot induce logical rules as explanations for predictions.
>
> It is worth noting that a possible way to apply KEPLER [1], PKGC[2] and SimKGC [3] to our scenario is first pre-training the PLMs such as BERT on the given corpus, and then utilizing the pre-trained PLM to address inductive KGC. Although this way does not require to process soft triples, it still fails to generalize in the real-world application scenarios. To analyze the reasons, we first recall the formalization of our problem setting.
>
> $\textbf{Problem statement}:$ Given a set of triples and a corpus for training $U\_{train}=(\mathcal{G}\_{train}, \mathcal{T}\_{train})$, a set of triples and a corpus for test $U_{test}=(\mathcal{G}\_{test}, \mathcal{T}\_{test})$, our inductive KGC setting aims to learn a KGC system based on $U_{train}$, and then evaluate the learnt system on $U_{test}$. During evaluation, for each head query $(?,r,t)$ or tail query $(h,r,?)$, the learnt system finds an answer with the highest estimated truth degree for answering $(?,r,t)$ or $(h,r,?)$, based on the background knowledge from $U_{train} \cup U_{test} \setminus \\{(\\{(h,r,t)\\}, \emptyset)\\}$.
>
> It can be seen in our inductive setting that the KGC systems require to process unseen text corpus in the test set. It is worth noting that this setting is motivated by the real-world application scenarios that we might need to fetch texts from search engines to find evidences for a new fact. PLM-based methods cannot work in this scenario because they can only obtain textual knowledge by pre-training the model on the given corpus. This process is impractical in the inference phase because the process of pre-training is time-consuming and requires massive computing resources. In contrast, the proposed LSTK-TELM can easily address the above scenario.
>
> Besides, we have tried to apply methods like KEPLER [1] to the second stage of LSTK but found it to be time-consuming. This is due to the extraction of over 100 million soft triples in certain datasets (e.g. the BioRel dataset), which leads to time-consuming processes for PLM-based methods to handle such a large volume of triples due to their extensive parameter sizes.
>
> We will include these discussions in the revised version.
>
> > [R3] Prompt-based KGC models can complete such triples based on text information similar to your work. Could you explain the difference between these models and LSTK?
>
> [A] The differences between the mentioned methods [1-3] and LSTK are two-fold. Firstly, the text knowledge considered in the work [1-3] is different from that in LSTK, where KEPLER [1], PKGC [2] and SimKGC [3] consider text information such as contexts of triples, descriptions aligned to entities (e.g., Wikidata5M) and pre-trained textual knowledge. In contrast, LSTK is proposed to address the KGC scenario where a part of the knowledge (i.e. real-world facts) exists in structured KG, while the other part exists in text corpus. The input of this setting is the given KG and its corresponding corpus. Thus no description is given to a entity or a fact. The textual knowledge considered in LSTK is a set of potential facts (i.e., the soft triples). Secondly, PLM-based methods such as [1-3] can also be seen as embedding-based methods, which are black-box models with low interpretability. In contrast, the proposed LSTK framework is able to induce logical rules (i.e. the TE-rules) as explanations for predictions.
>
> > [R4] I think you need to add that this approach is difficult to work on knowledge graphs that do not have corresponding text data to Limitations in your paper.
>
> [A] We have done this discussion in our supplement file (Please see section E: Discussion on Limitations).

---

> > ### Comment · Reviewer_Vj83 · 2023-08-14
> > **I appreciate your detailed explanations.**
> >
> > The authors' response has cleared the position of their work that attempts to accomplish interpretable KGC. Considering the possibility of revising the paper based on the response's content, I will increase my recommendation score.

---

> > > ### Author Response · Authors · 2023-08-15
> > > **Thanks for your helpful feedback!**
> > >
> > > Thank you for taking the time to review our paper and providing valuable comments. We will definitely improve our paper based on your comments. Please feel free to provide additional feedback and we will try our best to improve our work!

---

### Official Review · Reviewer_KZDY · 2023-07-07

**Soundness:** 2 fair
**Presentation:** 2 fair
**Contribution:** 2 fair
**Rating:** 5
**Confidence:** 4

**Summary:**

This paper proposes a rule-based inductive KGC method with two stages. In the first stage, it extracts soft triples from a text corpus using distant supervision. In the second stage, the obtained soft triples are mixed with the original hard triples and used to learn rule-based model for KGC.

**Strengths:**

(1) It is interesting and intuitive to enrich the original triples in KG by extract new triples from text corpus.
(2) The idea that uses extracted soft triples from text corpus to enhance logical rules is novel to some extent.

**Weaknesses:**

(1) This paper uses both structural and textual knowledge for inductive KGC, but there is no discussion about other related works which also use structural and textual knowledge, such as  KEPLER[1] and BERTRL[2].
(2) Since the proposed method extract new triples from text corpus, if the unseen entities in test set may seen in the new soft triples. It is better to provide statistics about the  unseen entities in original triples, soft triples and both triples.
(3) Some notations are confusing, for example the "t" is used for a tail entity in a triple (h,r,t), and also used in "1≤t≤T".

[1] Wang, X., Gao, T., Zhu, Z., Zhang, Z., Liu, Z., Li, J., & Tang, J. (2021). KEPLER: A Unified Model for Knowledge Embedding and Pre-trained Language Representation. Transactions of the Association for Computational Linguistics, 9, 176–194.
[2] [2] Zha, H., Chen, Z., & Yan, X. (2022). Inductive Relation Prediction by BERT. Proceedings of the AAAI Conference on Artificial Intelligence, 36(5), 5923-5931.

**Questions:**

(1) This paper uses both structural and textual knowledge for inductive KGC, but there is no discussion about other related works which also use structural and textual knowledge, such as  KEPLER[1] and BERTRL[2].
(2) Since the proposed method extract new triples from text corpus, if the unseen entities in test set may seen in the new soft triples. It is better to provide statistics about the  unseen entities in original triples, soft triples and both triples.
(3) Some notations are confusing, for example the "t" is used for a tail entity in a triple (h,r,t), and also used in "1≤t≤T".

[1] Wang, X., Gao, T., Zhu, Z., Zhang, Z., Liu, Z., Li, J., & Tang, J. (2021). KEPLER: A Unified Model for Knowledge Embedding and Pre-trained Language Representation. Transactions of the Association for Computational Linguistics, 9, 176–194.
[2] [2] Zha, H., Chen, Z., & Yan, X. (2022). Inductive Relation Prediction by BERT. Proceedings of the AAAI Conference on Artificial Intelligence, 36(5), 5923-5931.

**Limitations:**

(1) This paper uses both structural and textual knowledge for inductive KGC, but there is no discussion about other related works which also use structural and textual knowledge, such as  KEPLER[1] and BERTRL[2].
(2) Since the proposed method extract new triples from text corpus, if the unseen entities in test set may seen in the new soft triples. It is better to provide statistics about the  unseen entities in original triples, soft triples and both triples.
(3) Some notations are confusing, for example the "t" is used for a tail entity in a triple (h,r,t), and also used in "1≤t≤T".

[1] Wang, X., Gao, T., Zhu, Z., Zhang, Z., Liu, Z., Li, J., & Tang, J. (2021). KEPLER: A Unified Model for Knowledge Embedding and Pre-trained Language Representation. Transactions of the Association for Computational Linguistics, 9, 176–194.
[2] [2] Zha, H., Chen, Z., & Yan, X. (2022). Inductive Relation Prediction by BERT. Proceedings of the AAAI Conference on Artificial Intelligence, 36(5), 5923-5931.

---

> ### Author Rebuttal · Authors · 2023-08-08
>
> Thanks for your helpful comments. The following addresses concerns and questions.
>
> > [R1] This paper uses both structural and textual knowledge for inductive KGC, but there is no discussion about other related works which also use structural and textual knowledge, such as KEPLER[1] and BERTRL[2].
>
> Due to space limitations, we have only discussed the related work about learning rule-based systems for inductive KGC. We have not discussed the methods like KEPLER[1] and BERTRL[2] because they follow a different research line than our method. On one hand, these methods consider different textual knowledge from that of LSTK, where the textual knowledge considered in [1-2] encompasses the contexts of triples, descriptions on entities and the pre-trained knowledge. The purpose of introducing texts for these methods is to enrich the semantic information of entities, thereby obtaining improved entity representations. In contrast, we study the KGC setting where a part of the knowledge exists in the KG, while the other part (i.e. potential facts) exists in text corpus. In this setting, the input consists solely of the provided KG and its corresponding text corpus, without any accompanying entity descriptions. The textual knowledge considered in our work refers to the potential facts entailed by the text corpus. This setting is motivated by the scenario that the existing facts in structural KG  are highly incomplete, and we can fetch text evidences from text corpus to support a new fact. Therefore, we mine a set soft triples from the text corpus to obtain textual knowledge in the first stage.
> On the other hand, PLM-based methods such as KEPLER[1] and BERTRL[2] can be considered as embedding-based methods, which are black-box models with low interpretability, and cannot induce logical rules as explanations for predictions. In reality, we naturally expect that the proposed framework LSTK is highly interpretable to human. Thus, we focus on learning rule-based systems for inductive KGC because logical rules excel in explaining why a missing fact is inferred. We will include these discussions in the revised version.
>
> > [R2] Since the proposed method extract new triples from text corpus, if the unseen entities in test set may seen in the new soft triples. It is better to provide statistics about the unseen entities in original triples, soft triples and both triples.
>
> [A] Thanks for pointing out the statistic difference. We provide our statistical details as follow. The statistical results show that about 98.7%/83.7%/79.4% triples in the test set of HacRED/DocRED/BioRel involve unseen entities from the training triples, about 31.6%/84.9%/62.1% triples in the test set of HacRED/DocRED/BioRel involve unseen entities from the extracted soft triples, and about 30.0%/67.8%/46.8% triples in the test set of HacRED/DocRED/BioRel involve unseen entities from both the training triples and extracted soft triples. This shows that there are still a great number of triples in the test set involving unseen entities, even when the soft triples are used. Furthermore, we have done experiments to show whether the new extracted soft triples can cover the triples in test set (Please see the baseline Textual Copy Rule (TCR) and the variant model (2) in Table 2). We will include these statistics in the revised version.
>
>
> > [R3] Some notations are confusing, for example the "t" is used for a tail entity in a triple (h,r,t), and also used in "1≤t≤T".
>
> [A] Thank you for pointing out it. We will refine this notation in the revised version.

---

> > ### Comment · Reviewer_KZDY · 2023-08-16
> >
> > The authors' response has cleared some issues, I will increase my recommendation score.

---

> > > ### Author Response · Authors · 2023-08-16
> > > **Thanks for your helpful feedback!**
> > >
> > > Thank you for taking the time to review our paper and providing valuable comments. We will definitely improve our paper based on your comments. Please feel free to provide additional feedback and we will try our best to improve our work. Thank you for the helpful discussion for improving our work again!

---

### Official Review · Reviewer_DS3c · 2023-07-11

**Soundness:** 2 fair
**Presentation:** 3 good
**Contribution:** 2 fair
**Rating:** 5
**Confidence:** 4

**Summary:**

This paper addresses the problem of knowledge graph completion by proposing a two-stage framework — learning from structural and textual knowledge (LSTK). This novel framework leverages both structural and textual knowledge to learn rule-based systems, which provides a unique approach in the realm of KGC research. The paper is premised on the idea that the typical reliance on structural knowledge alone in rule-based systems is a limiting factor. This leads to the proposition of a system that utilizes "soft triples", or triples with confidence scores derived from textual corpora via distant supervision and a textual entailment model with multi-instance learning. In the second stage, a rule-based system for KGC is learned using both these soft triples and the hard triples of existing KG triples. The paper also introduces a novel formalism for learning rules, referred to as text-enhanced rules or TE-rules, which can mitigate the negative impact of noisy data in soft triples.


**Strengths:**

— The two-stage framework introduces an innovative way of leveraging both structural and textual knowledge for KGC, thereby potentially mitigating the limitations of existing methods.

— The introduction of TE-rules provides a robust mechanism for learning rules, to mitigate the negative impact of noises from soft triples.

— The paper introduces three new datasets for empirical evaluation, contributing further to the body of resources available in this field.


**Weaknesses:**

— The authors utilize a single method to estimate the confidence scores of the soft triples, resulting in a potential lack of generalizability.

— While the introduction of three new datasets is commendable, the paper does not offer a clear comparison of these datasets with established benchmarks.

— Furthermore, it remains uncertain how well the proposed method would perform on existing benchmarks.

— The paper falls short in discussing the scalability of the proposed method. It is unclear how the proposed framework would handle large-scale knowledge graphs, particularly considering the potential computational costs of generating and handling soft triples.

**Questions:**

— Can you elaborate why only use a textual entailment model based on the pre-trained language model BERT to estimate the confidence scores of the soft triples and discuss its generalizability? And what exact BERT model variant do authors use?

— How do the newly introduced datasets compare with established benchmarks? Moreover, how would the proposed LSTK framework perform on these existing benchmarks?

---

> ### Author Rebuttal · Authors · 2023-08-08
>
> Thanks for your helpful comments. The following addresses concerns and questions.
>
> > [R1] The authors utilize a single method to estimate the confidence scores of the soft triples, resulting in a potential lack of generalizability.
>
> [A] We have conducted an ablation study on replacing the proposed method for extracting soft triples (please see variant model (6) in Table 2). This variant model employs a strong baseline [1] in the field of distantly supervised relation extraction. Results show that the proposed method outperforms the variant in three datasets. This demonstrates the generalizability of the textual entailment model.
>
> > [R2] While the introduction of three new datasets is commendable, the paper does not offer a clear comparison of these datasets with established benchmarks.
>
> [A] As far as we know, there are no established benchmarks with corresponding text corpora that meet our settings. In our settings, we assume that a part of the knowledge (i.e., real-world facts) exists in structured KG, while the other part (i.e., potential facts) exists in text corpus. Consequently, we introduce three new datasets to evaluate the effectiveness of LSTK and facilitate the research field of addressing KGC from both structural and textual knowledge perspectives. The most relevant dataset for our study is the Wikidata5M [2] dataset, which provides aligned text descriptions on entities. While this dataset can be utilized to assess approaches focused on enhancing entity embeddings for KGC, it is not suitable to be used in our scenario, as our scenario assumes that a set of potential facts is implied by the corpus, and we require to mine these potential facts from the given corpus. Furthermore, our scenario presents a greater challenge since the relationship between entities and the provided text corpus is unknown, whereas Wikidata5M provides aligned descriptions for each entities. We will include these discussions in the revised version.
>
>
> > [R3] Furthermore, it remains uncertain how well the proposed method would perform on existing benchmarks.
>
> [A] As mentioned in our discussion on limitations (Please see section E: Discussion on Limitations in the supplement file), our proposed framework, including the proposed TE-rules and TELM model, cannot effectively function for KGs without text corpora. For example, the introduced textual relations in TE-rules exist exclusively within KGs with soft triples. Therefore, we have not conducted an evaluation of LSTK-TELM on established benchmarks such as WN18RR, FB15k-237 and YAGO3-10.
>
> > [R4] The paper falls short in discussing the scalability of the proposed method. It is unclear how the proposed framework would handle large-scale knowledge graphs, particularly considering the potential computational costs of generating and handling soft triples.
>
> [A] As reported in Table 1, there are about 38.8/4.7/123.4 million soft triples has been extracted in the first stage for the HacRED/DocRED/BioRel datasets. The computational cost of generating soft triples is mainly on the textual entailment model based on BERT, where it takes about 6/0.2/18 hours to compute all soft triples for the HacRED/DocRED/BioRel datasets. Further, the proposed LSTK-TELM model is able to handle both the existing triples and soft triples in efficient time, as it stores all the triples by sparse adjacency matrix. In more details, each query in the HacRED/DocRED/BioRel datasets takes about 1.5/0.3/2.5 seconds for evaluation. We will include these discussions in the revised version.
>
>
> > [R5] Can you elaborate why only use a textual entailment model based on the pre-trained language model BERT to estimate the confidence scores of the soft triples and discuss its generalizability? And what exact BERT model variant do authors use?
>
> [A] We use the BERT-base model for textual entailment. Implementation details about the textual entailment have been reported in section 5.2. Actually, we have discussed the generalizability of the textual entailment model in Table 2. Specifically, we have compared with a strong baseline [1] in the field of distantly supervised relation extraction. (please see variant model (6) in Table 2). Results show that the proposed method outperforms this variant in three datasets. This demonstrates the generalizability of the textual entailment model. This may be due to that a textual entailment model can deal with open relations and exploit more text semantics (e.g., contexts of relations) on relations than a relation extraction model does (please see line 311-315 for more details).
>
> > [R6] How do the newly introduced datasets compare with established benchmarks? Moreover, how would the proposed LSTK framework perform on these existing benchmarks
>
> [A] As far as we know, there are no established benchmark KGs with corresponding text corpora that meet our problem settings, thus we introduce three new datasets for evaluation. The most relevant dataset for our study is the Wikidata5M [2] dataset, which offers aligned text descriptions on entities. Nevertheless, Wikidata5M is not suitable to be used in our scenario because its text corpus mainly encompasses descriptions on entities, whereas our scenario requires extracting a set of potential facts from the given corpus. Therefore, we have not evaluated LSTK on Wikidata5M.
>
> On the other hand, the proposed TE-rules and TELM model are designed to handle soft triples with textual relations. Thus, we have not evaluated LSTK-TELM on existing benchmarks such as WN18RR, FB15k-237 and YAGO3-10. This has also been discussed in the section on limitations (please see section E: Discussion on Limitations in the supplement file).
>
> **Reference:**
>
> [1] Fine-tuning pre-trained transformer language models to distantly supervised relation extraction. In ACL, pages 1388–1398, 2019.
>
> [2] KEPLER: A Unified Model for Knowledge Embedding and Pre-trained Language Representation. In TACL, 9, 176–194, 2021.

---

> > ### Comment · Reviewer_DS3c · 2023-08-15
> > **Thanks for your response!**
> >
> > I appreciate the author's rebuttal, and most of my concerns have been addressed. So, I am inclined to accept this paper.

---

> > > ### Author Response · Authors · 2023-08-16
> > > **Thanks for your helpful feedback!**
> > >
> > > Thank you for taking the time to review our paper and providing helpful comments. We will definitely improve our paper based on your feedback. Please feel free to provide additional feedback and we will try our best to improve our work. Thanks again!

---

### Decision · Program_Chairs · 2023-09-21

**Decision:**

Accept (poster)

**Comment:**

In this paper, the authors address knowledge graph completion (KBC) by introducing a two-stage framework that incorporates both structural and textual knowledge. They first generate "soft triples" with confidence scores from text data using distant supervision, using a multi-instance learning model to determine if a triple is entailed by sentences. In the second stage, they use these soft triples to learn rule-based models for KBC. To handle noise in soft triples, they propose "text-enhanced rules" (TE-rules) and a neural model to simulate their reasoning process. The approach is evaluated on three benchmark datasets, outperforming baseline methods in terms of various metrics such as MRR and Hits@k. The paper introduces an innovative way to leverage both structured and textual information for inductive KGC.